# Elastic anisotropies of deformed upper crustal rocks in the Alps

Ruth Keppler (1), Roman Vasin (2), Michael Stipp, (3), Tomás Lokajícek (4), Matej Petruzálek (4), Nikolaus Froitzheim (1)

Corresponding author: Ruth Keppler (rkep@uni-bonn.de)

[1] Institute for Geosciences, University of Bonn, Poppelsdorfer Schloss, D-53115 Bonn, Germany

[2] Frank Laboratory of Neutron Physics, Joint Institute for Nuclear Research, Joliot-Curie 6, 141980 Dubna, Russia

[3] Institute for Geosciences and Geography, Von-Seckendorff-Platz 3, D-06120 Halle (Saale), Germany

[4] Institute of Geology of the Czech Academy of Sciences, Rozvojova 269, 16000 Prague 6, Czech Republic

ABSTRACT

The crust within collisional orogens is very heterogeneous both in composition and grade of deformation, leading to highly variable physical properties at small scales. This causes difficulties for seismic investigations of tectonic structures at depth since the diverse and partially strong upper crustal anisotropy might overprint the signal of deeper anisotropic structures in the mantle. In this study, we characterize the range of elastic anisotropies of deformed crustal rocks in the Alps. Furthermore, we model average elastic anisotropies of these rocks and their changes with increasing depth due to the closure of microcracks. For that pre-Alpine upper crustal rocks of the Adula Nappe in the central Alps, which were intensely deformed during the Alpine orogeny, were sampled. The two major rock types found are orthogneisses and paragneisses, however, small lenses of metabasites and marbles also occur. Crystallographic preferred orientations (CPOs) and volume fractions of minerals in the samples were measured using time-of-flight neutron diffraction. Combined with single crystal elastic anisotropies these were used to model seismic properties of the rocks. The sample set shows a wide range of different seismic velocity patterns even within the same lithology, due to the microstructural heterogeneity of the deformed crustal rocks. To approximate an average for these crustal units, we picked common CPO types of rock forming minerals within gneiss samples representing the most common lithology. These data were used to determine an average elastic anisotropy of a typical crustal rock within the Alps. Average mineral volume percentages within the gneiss samples were used for the calculation. In addition, ultrasonic anisotropy measurements of the samples at increasing confining pressures were performed. These measurements, as well as the microcrack patterns determined in thin sections were used to model the closure of microcracks in the average sample at increasing depth. Microcracks are closed at approximately 740 MPa yielding average elastic anisotropies of 4% for the average gneiss. This value is an approximation, which can be used for seismic models at a lithospheric scale. At a crustal or smaller scale, however local variations in lithology and deformation as displayed by the range of elastic anisotropies within the sample set need to be considered. In addition, larger scale structural anisotropies such as layering, intrusions, as well as brittle faults have to be included in any crustal scale seismic model.

## 1. Introduction
Geophysical studies of the Earth's crust and mantle are continuously improving allowing for more and
more detailed structural investigationsdue to higher resolutions at increasingly greater depth. High-
resolution geophysical imaging of 3D structures is currently carried out within the AlpArray initiative using
a high-end seismological array in the Alpine orogeny (Hetényi et al., 2018). For this as well as other similar
projects around the world precise knowledge of the physical properties of the rocks at depth is required.
Especially elastic anisotropy data are of importance, since they reflect shearing at depth. Elastic anisotropy
of mantle rocks is in large parts caused by the crystallographic preferred orientation (CPO) of the
constituent mineral phases (Silver, 1996; Montagner and Guillot, 2003). Besides CPO other rock fabrics
such as compositional layering, grain and aggregate size and shape, grain boundaries and shape preferred
orientation can bear an influence. At shallower depth microcracks additionally modify elastic properties
by both lowering the seismic velocity and increasing the elastic anisotropy in deformed rocks. The elastic
rock properties can be either be gained by measurements using ultrasound, including experiments at high
pressures and temperatures (e.g., Christensen, 1965; Babuška, 1968; Christensen, 1979; Christensen and
Mooney, 1995; Kern and Wenk, 1990; Pros et al., 2003), or modeled using the CPO data of the constituent
minerals and their corresponding single crystal elastic anisotropies (e.g., Mainprice and Humbert, 1994;
Bascou et al., 2001; Cholach and Schmitt, 2006; Llana-Fúnez and Brown, 2012; Almqvist and Mainprice,
2017; Puelles et al., 2018). Many works combine these two approaches to highlight the effect of individual
minerals on elastic wave velocities in bulk rock, or to infer the influence of pores and fractures (e.g., Ji and
Salisbury, 1993; Ji et al., 1993; Barruol and Kern, 1996; Mauler et al., 2000; Ji et al., 2003; Ivankina et al.,
2005; Kitamura, 2006; Kern et al., 2008; Ábalos et al., 2010; Lokajicek et al., 2014; Keppler et al., 2015;
Vasin et al.,2017; Ullemeyer et al., 2018). During experimental measurements, microcracks in rock samples
are not completely closed, despite pressure vessels operating at up to hundreds of MPa during
measurements (e.g. Christensen, 1974; Kern et al., 2008; Matthies, 2012; Vasin et al., 2017). That is why
resulting data are only comparable to elastic anisotropiesof crustal depth, whereas the modeled
anisotropies yield results for a crack free medium at higher depths (e.g., within thickened crust or at mantle
depth).
When using elastic anisotropy data of natural rocks as input parameters for seismic investigation the gap
between the km-scale of detectable units in seismic imaging at depth and the centimeter-sized rock
samples taken from outcrops in meter scales must be considered. This difference in scale is less
problematic for the relatively homogenous mantle rocks with a fairly simple mineralogy (e.g. Mainprice et
al., 2000; Karato et al., 2008), but even in the mantle compositional heterogeneities leading to elastic
anisotropies have been observed (Faccenda et al., 2019). Crustal rocks are not only polymineralic but
lithologies significantly vary in composition. Additionally, deformation is also very heterogeneous within
the crust. Especially subduction zones and collisional orogens show a complex deformational history (e.g.,
Schmid et al., 2004; Simancas et al., 2005; Zhang et al., 2012). This results in a large variety of CPO patterns
throughout a kilometer scale geological unit (Schmidtke et al. 2021). Averaging the calculated or measured
elastic anisotropies may lead to the assumption of an unrealistically isotropic medium, for these strongly
deformed parts of the crust. There are only a few studies, which aim to close the gap between the elastic
anisotropy gained from hand samples-sized volumes and the one measured in seismic experiments of the
crust and mantle (Okaya et al., 2019; Zertani et al., 2020). Okaya et al. (2019) investigated the influence of
local structures such as folds, domes or shear zones on the bulk anisotropic properties of larger units.
Using tensor algebra they separate these local structures from an already overall anisotropic rock, which
allows to quantify the role of macroscale structures. Zertani et al. (2020) used the finite element method
to model petrophysical properties of meter to kilometer scale eclogite units, which could allow to visualize
structures in active subduction and collision zones by geophysical methods.
In the present work, we classify the crust according to its composition and grade of deformation in order
to define larger units which can be summarized. Since only deformed parts of the crust exhibit elastic
anisotropy, this study is focused on the Adula Nappe of the Central Alps. Originating from pre-Alpine upper
crust mainly made up of granitoids and Mesozoic sediments, the Adula Nappe was intensely deformed
during the Alpine Orogeny. CPO as well as volume percentages of all mineral phases from a large set of
samples of this unit were determined. Subsequently, elastic anisotropies of the samples were calculated.
These show a wide range of seismic properties of deformed crustal rocks in the Alps. Most of the samples
are gneisses, which represent the most common rock type in the Adula Nappe. Based on the characteristic
CPO types, average CPO strengths and average volume percentages of the relevant mineral phases, we
calculated the elastic anisotropy of an "average rock", which represents an average anisotropy for
deformed crustal rocks in collisional orogens. The two major lithologies are orthogneisses and
paragneisses, which is why the "average rock" has typical gneiss CPO and composition. Because of the
importance of microcracks at shallow depth, we used data from ultrasonic measurements as well as thin
section analysis to determine typical crack patterns in the samples. From these the influence of
microcracks on elastic properties was quantified, as well as the changes in elastic anisotropy with
increasing depth up to the point where all microcracks are presumably closed.
This is, of course, a simplification of the very heterogeneous crust of the Alps, as already shown by the
variability of elastic anisotropy of the individual samples from the Adula Nappe. Yet, such an average rock
can be used for lithospheric and upper mantle scale seismic models, in which the crust is implemented as
a single unit with an average anisotropy. At crustal scale the heterogeneity of different rocks caused by
variable composition as well as variable deformation have to be considered. While it is difficult to present
a universal average anisotropy for the very heterogeneous crust within collisional orogens, this
contribution aims to bridge the scale gap between elastic anisotropy data of rock samples and the
kilometer scale structures measured in seismic investigations by considering heterogeneities in
composition and structure as well as the reduction of crack porosity with increasing depth.
2. Elastic anisotropies within the Alpine orogen
The Alpine orogen exhibits a mountain-belt-parallel seismic anisotropy (e.g., Silver, 1996; Smith and
Ekström, 1999; Bokelmann et al., 2013; Petrescu et al., 2020), which is not completely understood. In the
Western Alps this anisotropy was illustrated by teleseismic shear wave splitting and interpreted as a result
of asthenospheric flow beneath the lithospheric slab, although a further influence by lithospheric
anisotropy due to Alpine deformation could not be excluded (Barruol et al., 2004; 2011). Fry et al. (2010),
on the other hand, determined seismic anisotropies within the Alps by passive seismic imaging using
Rayleigh wave phase velocities. Their results suggest two distinct vertically distributed layers of anisotropy
– an orogen-parallel fast direction down to 30 km and an orogen-perpendicular one between 30 and 70
km depth - with differing geodynamic origins. The authors interpret the orogen-parallel anisotropy as a
consequence of the CPO of crustal minerals (e.g. amphibole and biotite) in response to compression and
consider the deeper, orogen-perpendicular anisotropy to result from bending and flow of the European
lithospheric mantle. This two-layer anisotropy was also detected from SKS-splitting in the transition to the
Eastern Alps. The two layers were interpreted as asthenospheric flow above a detached lithospheric slab
fragment with mountain chain parallel CPO (Qorbani et al., 2015; Link and Rümpker, 2021).
The Alps have a fairly complicated tectonic history with two major collisional events involving several
oceans and microcontinents. While the cretaceous Eoalpine event only involved the Eastern Alps, the
Tertiary deformation incorporated the complete Alpine orogen. Here, we concentrate on the deep
structure of the Western and Central Alps that mainly result from Paleogene and Neogene tectonics when
the Penninic ocean basins were subducted and Adria, Iberia, and other continental fragments collided with
Europe. We consider a simplified version of the NFP-20 EAST&EGT profile (Fig. 1A; Schmid and Kissling,
2000) ande exclude nappe structures in the shallowest part of the profile, like the Helvetic nappes. This
results in a profile including the following upper crustal units: the Aar and Gotthard massifs representing
weakly deformed European basement; the Lucomagno, Simano and Adula nappes of deformed European
basement and Mesozoic cover; and relatively undeformed Apulian upper crust. To simplify, we therefore
subdivide the profile into
(1) weakly deformed and isotropic upper crust
(2) strongly deformed anisotropic upper crust mostly comprising gneiss (Fig. 1B).

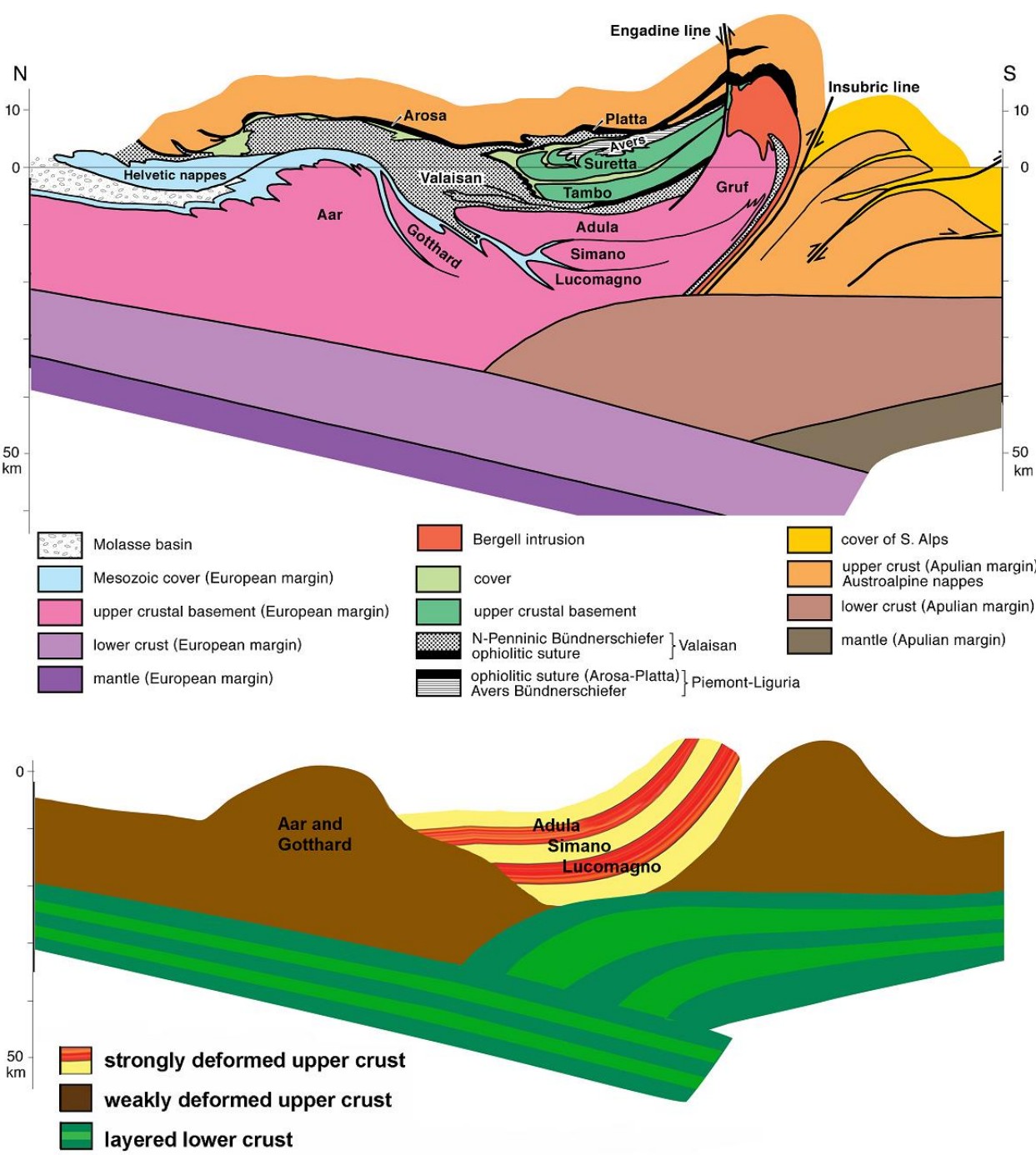


Figure 1: (A) North-south tectonic profile through the central Alps showing all major units (NFP-20 EAST&EGT; Schmid and Kissling, 2000) (B) strongly simplified profile consisting of the predominant rock units and neglecting the sedimentary cover and ophiolite units


## 2.1. Weakly deformed Alpine upper crust


In this study, both the crystalline massifs in the northern part of the central Alps and the Adriatic basement in the Southern Alps are assumed to show weak or no elastic anisotropy.

The Aar and Gotthard massifs contain large Variscan granitoid bodies which intruded into a pre-Variscan
basement. These units were only weakly overprinted by Alpine metamorphism and deformation (e.g.,
Abrecht, 1994; Schaltegger, 1994; Oliot et al., 2010). However, some greenschist to amphibolite facies
shear zones have been documented, which have to be considered for any large scale model (Challandes
et al., 2008; Goncalves et al., 2012; Wehrens et al., 2017). In addition structures related to the evolution
of Gondwana in the pre-Variscan basement, in which the granitoids intruded also have to be regarded (e.g.
von Raumer et al., 2013). Furthermore, Jurassic rifting structures are present in parts of the Penninic
nappes (e.g. Froitzheim and Manatschal, 1996). Even though these structures are mostly related to brittle
deformation, they might cause local seismic anisotropies.
In the Southern Alps, metamorphic grade during deformation was generally low. Deformation in the
basement is limited to large scale thrust faults during Alpine tectonics (e.g., Laubscher 1985). For
simplification, we are assuming an elastically isotropic medium for both the Aar and Gotthard massifs of
the European margin and the Southern Alps due to the lack of pervasive CPO forming deformation.
However, local ductile shear zones as well as large brittle faults also have an influence on the overall elastic
anisotropy (e.g. Almqvist et al., 2013).
Of course one needs to bear in mind that considering the crystalline massifs in the northern part of the
central Alps and the Adriatic basement as isotropic is a strong simplification of complex structures with a
long deformational history. In addition to brittle deformation structures, lithological layering as well as
intrusions may be further factors influencing the overall anisotropy of crustal scale seismic models.
2.2. Strongly deformed Alpine upper crust
As indicated by numerous geological field studies as well as strong reflectors in the original NFP-20-east
seismic profile (Pfiffner et al., 1988) , the crustal units in the main part (concerning their position in the N-
S running profile) of the central Alps have been strongly deformed during subduction and subsequent
continental collision (Fig. 1B).
The Adula Nappe together with the Simano and Lucomagno nappes constitutes the Lepontine dome,
which mostly consists of Alpine nappes including Variscan basement and its Mesozoic cover (e.g. Engi et
al., 1995; Nagel et al., 2002). In this study, the Adula Nappe is taken as an example for the strongly
deformed parts of the Alps, representing a relatively coherent unit with stratigraphic basement-cover
contacts. It comprises orthogneisses from Cambrian, Ordovician, and Permian protoliths (Cavargna-Sani
et al. 2014), paragneises with metabasic lenses, and some layers of marble (Fig. 2). It was originally part of
the distal European continental margin and entered a south-dipping subduction zone in which the Valais
(North Peninnic) Ocean had been consumed. The unit shows peak conditions of 12–17 kbar/500–600 C° in
the north and 30 kbar/800–850 C° in the south (e.g. Heinrich, 1986; Löw,1987; Meyre et al., 1997; Nagel
et al. 2002; Dale and Holland, 2003). Lu–Hf garnet ages revealed an Eocene age for UHP metamorphism
(35–38 Ma; Sandmann et al., 2014) All lithologies found in the nappe were sampled, however most
samples are orthogneisses and paragneisses, since these lithologies make up the largest part of the nappe
and other lithologies might be too small scale to be detected in seismic imaging. However, since these
layers of different lithology could be significant for the overall anisotropy two metabasalts as well as a
marble sample have been included in the sample set.

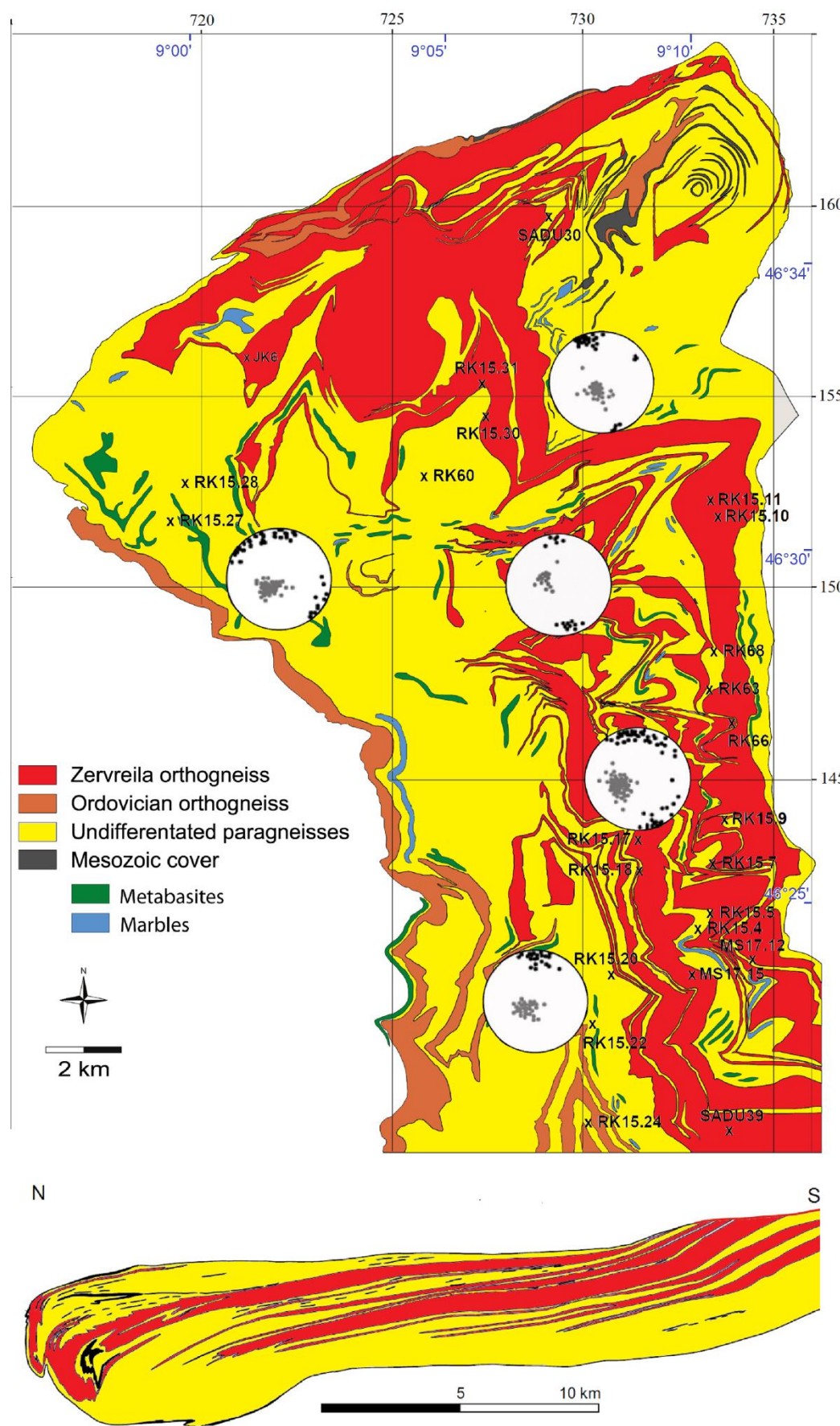

Figure 2: Simplified tectonic map and north-south profile (along 730 line of longitude) of the Adula Nappe
(modified after Nagel. 2008 and Cavargna-Sani et al., 2014). Grey and black dots indicate poles of main
foliation and stretching lineation, respectively, of the central Adula Nappe. Sample locations are indicated.
Swiss coordinates are marked in black; UTM coordinates are marked in blue.
From peak conditions to its current position within the Lepontine dome, the Adula Nappe underwent
several deformation phases. The oldest, peak to post-peak deformation phase is the eclogite facies
Zapport phase, which is well documented in the central part of the nappe, where it was not overprinted
by younger deformation phases (e.g. Löw, 1987; Meyre et al., 1993; Pleuger et al., 2003). The Zapport
phase records the earliest stages of exhumation and led to boudinage of the eclogite lenses, isoclinal
folding, an axial plane foliation, a N-S-trending stretching lineation, as well as a top-to-the-north sense of
shear (Meyre et al., 1993). Samples used for this study are from this area and represent deformed crustal
parts of the Alps.
3. Methods
3.1. CPO analysis
CPO measurements were performed at the neutron time-of-flight (TOF) texture diffractometer SKAT at
the Frank Laboratory of Neutron Physics at JINR, Dubna, Russia (Ullemeyer et al., 1998; Keppler et al.,
2014). The high penetration capability of neutrons into matter together with the large beam cross section
of the SKAT (50 x 95 mm$^2$) allow measurements of large-volume samples. In this study, roughly spherical
samples with volumes of about 65 cm$^3$ were measured. Since the investigated samples are usually coarse-
grained this guarantees good grain statistics. Moreover, since diffraction patterns are recorded in a TOF
experiment over a large interval of lattice spacings, often containing hundreds of diffraction peaks, the so-
called 'Rietveld Texture Analysis' can be used for the texture evaluation, allowing the simultaneous
determination of all mineral textures even for samples with complex mineralogy (Von Dreele, 1997;
Matthies et al. 1997), as well as defining the rock mineral composition. We used the MAUD software for
the texture evaluation (Lutterotti et al., 1997; Wenk et al., 2010; Schmidtke et al., 2021). For every sample,
a sample coordinate system XYZ representing the three directions of the finite strain ellipsoid was chosen.
X is the lineation direction, Y is within the foliation plane perpendicular to the lineation and Z is the foliation
normal.
3.2. Modeling of elastic anisotropies
From the orientation distribution function (ODF) of the main rock constituents, their volume fractions in
each sample and particular single crystal elastic constants, the elastic moduli of bulk rock were calculated
using the BEARTEX software (Wenk et al., 1998). For that purpose, averaging schemes are often used, such
as Voigt approach (Voigt, 1887) or Reuss approach (Reuss, 1929). The former assumes that all crystallites
in the polycrystal are under the same strain, while the latter considers equal stress state in all crystallites.
To get a first approximation on the different elastic anisotropy patterns within the set of samples, we used
the Voigt averaging scheme that provides reasonably good agreement of rock petrofabric data and
laboratory measurements (Ben Ismail and Mainprice, 1998), while noting that the recalculated elastic
properties represent the upper boundary of the polycrystal stiffness.
The single crystal elastic constants for the calculation were taken from the literature (muscovite: Vaughan
and Guggenheim, 1986; quartz: Heyliger et al., 2003; albite: Brown et al., 2006; calcite: Dandekar, 1968;
dolomite: Humbert & Plique, 1972; hornblende: Aleksandrov and Ryzhova, 1961; epidote: Aleksandrov et
al., 1974; garnet: Zhang et al., 2008; omphacite: Bhagat et al., 1992). Phase elastic wave velocities were
calculated from bulk elastic tensors of rocks using the Christoffel equation.
To calculate the elastic anisotropy of the "average rock", representative of crustal lithology, and its
changes with overburden depth due to closure of the microcracks (see section 4.5), a more sophisticated
approach to the calculation of rock elastic properties is necessary. We used a modified self-consistent
method GeoMIXself (GMS; Matthies, 2010; 2012), which combines the standard self-consistent routines
(e.g. Morris 1970) with elements of the geometric mean averaging (Matthies & Humbert 1995). This
method is able to take CPO, morphologies and shape preferred orientations (SPOs) of grains, as well as
pores and cracks, into account. Similar to self-consistent approach, in GMS all rock constituents (mineral
grains, pores or microcracks) are approximated by oblate spheroids. Details and limitations of this
approach for an application to polymineral rocks are discussed in, e.g. Vasin et al. (2013), Vasin et al. (2017)
and Lokajicek et al. (2021).
3.3. Ultrasonic measurements
From the sample set, two samples with common CPO patterns and strengths of their constituent mineral
phases were picked for ultrasonic measurements of P-wave velocity distributions at the pressure
apparatus of the Institute of Geology ASCR, Prague, Czech Republic (e.g. Lokajicek et al., 2014). The
measurements were conducted on spherical samples with diameters of 41.0 mm (RK15-17) and 39.4 mm
(RK15-22), respectively. Before the measurement, the samples were dried at 100°C for 24 hours.
Afterwards they were covered by a thin layer of epoxy resin to protect inner pore space of the sample
against the hydrostatic pressure. Transformer oil served as the hydraulic medium. Ultrasonic signals were
excited and recorded using a pair of piezoceramic sensors with a resonant frequency of 2 MHz. P-wave
velocities were measured during loading in 132 independent directions at differing confining pressure
levels from ambient conditions to a maximum pressure of 300 or 400 MPa.
4. Sample description
The orthogneiss samples consist of quartz, plagioclase, kalifeldspar and mica (Table 1A). Mica is mostly
white mica but a few samples also contain biotite. Mica is frequently aligned within the foliation plane. It
occurs in layers in some samples but exhibits single grains or clusters scattered within a matrix of quartz
and feldspar in most orthogneisses. Microcracks in mica grains are mostly aligned with its basal plane,
however there are also some mircocracks cutting across basal planes (Fig 3A-C). Quartz  exhibits the full
range of dynamic recrystallization microstructures from grain boundary migration to subgrain rotation
recrystallization and bulging.

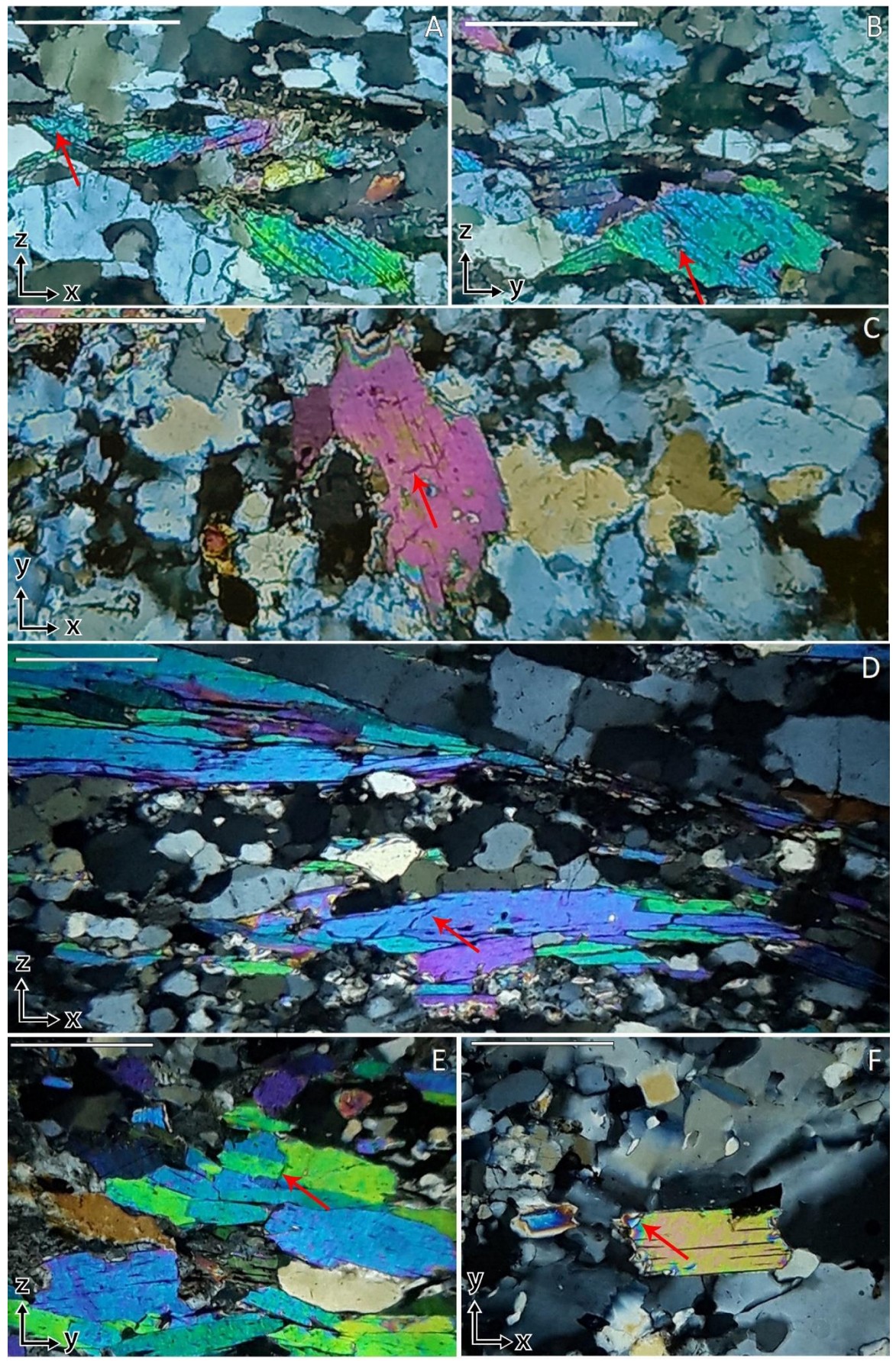

Figure 3: Thin sections of RK15-17, a typical orthogneiss (A, B, C) and RK15-22, a typical paragneiss (D, E,
F) under crossed polarizers for the XZ (A, D), YZ (B, E) and XY plane (C, F), showing examples microcracks
(red arrows) in mica. Arrows in left corner indicate the three directions of the finite strain ellipsoid. White
bar on upper left corner in each picture shows the length of 500 µm.

| A | Composition in Volume % | Location | B | Composition in Volume % | Location |
|---|---|---|---|---|---|
| GAN12 | 48 Qz, 31 Plg, 21 Kfs | Alp de Ganan | GAN08 | 36 Qz, 23 Plg, 31 Mica, 10 Grt | Alp de Ganan |
| JK6 | 39 Qz, 43 Plg, 18 Mica | 720 652/155 999 | GAN15 | 45 Qz, 26 Plg, 29 Mica | Alp de Ganan |
| MS17-15 | 25 Qz, 32 Plg, 11 Kfs, 32 Mica | 732 692/140 078 | MS17-12B | 51 Qz, 20 Plg, 29 Mica | 734 127/140 223 |
| RK15-9A | 71 Qz, 9 Plg, 20 Kfs | 732 876/144 686 | MS17-12C | 32 Qz, 42 Plg, 26 Hlb | 734 127/140 223 |
| RK15-9B | 60 Qz, 24 Plg, 15 Kfs, 1 Mica | 732 876/144 686 | RK15-5 | 60 Qz, 25 Plg, 15 Mica | 732 933/142 432 |
| RK15-10 | 71 Qz, 19 Plg, 10 Mica | 733 398/151 952 | RK15-18 | 16 Qz, 28 Plg, 56 Mica | 730 110/142 903 |
| RK15-11A | 33 Qz, 32 Pl, 35 Kfs | 722 272/152 194 | RK15-22 | 55 Qz, 15 Plg, 30 Mica | 729 771/139 042 |
| RK15-17 | 35 Qz, 43 Plg, 22 Mica | 729 661/143 839 | RK60 | 25 Qz, 70 Plg, 5 Cc | 726 875/152 275 |
| RK15-20 | 50 Qz, 41 Plg, 9 Mica | 730 265/140 481 | RK68 | 50 Cc, 50 Dol | 732 536/149 964 |
| RK15-24B | 38 Qz, 52 Plg, 14 Mica | 730 008/136 819 | RK70A | 36 Qz, 38 Plg, 26 Mica | 737 323/136 241 |
| RK15-27B | 63 Qz, 37 Plg | 719 193/152 476 | SADU16 | 42 Qz, 10 Plg 43 Mica, 5 Grt | 732 641/134 758 |
| RK15-28 | 34 Qz, 52 Plg, 14 Mica | 719 424/153 347 | SADU30 | 41 Qz, 25 Plg, 34 Mica | 731 985/162 618 |
| RK15-30B | 29 Qz, 60 Plg, 11 Mica | 727 713/156 013 | ZAP01 | 29 Qz, 23 Plg, 37 Mica, 7 Grt, 4 Hbl | near Zapporthütte |
| RK15-31 | 76 Qz, 4 Plg, 20 Kfs | 727 713/156 835 | | | |
| RK63B | 35 Qz, 32 Plg, 33 Kfs | 731 539/148 966 | C | Composition in Volume % | Location |
| RK66 | 37 Qz, 33 Plg, 30 Kfs | 732 554/148 402 | RK15-4 | 7 Qz, 29 Plg, 53 Hbl, 11 Omp | 732 078/141 893 |
| SADU39 | 58 Qz, 25 Plg, 17 Mica | 733 687/139 694 | RK15-7 | 15 Qz, 31 Plg, 51 Hbl, 3 Czo | 732 467/143 492 |

Table 1: Sample locations in Swiss coordinates and mineral volume percentages of (A) orthogneisses, (B)
paragneisses and (C) metabasites. Cc: calcite, Czo: clinozoisite, Dol: dolomite, Grt: garnet, Hbl: hornblende,
Kfs: kalifeldspar, Omp: omphacite, Plg: plagioclase, Qz: quartz.

The mineral compositions of the paragneisses is more variable. Similar to the orthogneiss samples, the
paragneisses consist of quartz, plagioclase and mica, however, there is no kalifeldspar in the samples and
the mica contents are generally higher (Table 1B). A few samples (RK15-18, SADU16) have a high mica
content of up to 56% and are therefore correctly termed mica schists. As they fall into the same category
of clastic metasediments, they are counted among the paragneisses which are the predominant rock type
of that group. They were also considered for the calculation of the average sample, concerning
composition and CPO. White mica is more common in the paragneisses than in the orthogneisses.
However, even biotite occurs more frequently in the paragneisses. One of the paragneiss samples contains
hornblende and several of the samples contain garnet. Mica appears more frequently aligned in layers
compared to the orthogneisses. Microcracks are mostly parallel to the mica basal plane with some
exceptions (Fig. 3D-F). Quartz microstructures also correspond to those of the orthogneisses.
The marble sample comprises equal amounts of calcite and dolomite, both of which exhibit an SPO with
an alignment in the foliation. The metabasites are strongly retrogressed eclogites consisting of about 50%
hornblende and variable amounts of quartz, plagioclase, omphacite and clinozoisite (Table 1C).
Hornblende shows an alignment within the foliation plane and is preferentially oriented parallel to the
stretching lineation.

5. Results

5.1. Crystallographic preferred orientation

Within the gneiss samples two major CPO patterns occur for quartz. In the first, quartz (0001) yields a
maximum between the Z- and Y-directions of the pole figure. This pattern occurs in 55% of the samples
containing quartz. In the second pattern, quartz (0001) exhibits peripheral maxima at an angle to the
foliation normal, occurring in 45% of the samples (Fig. 4A and APPENDIX - Table A1). Both fabrics can
contain subordinate girdle distributions. Similar quartz (0001) fabrics have been described for other high
pressure gneiss samples (e.g. Kurz et al., 2002; Keller and Stipp, 2011; Keppler et al., 2015). Although the
two patterns occur throughout the sample set, the former is more common in the paragneisses, while the
latter occurs more frequently in the orthogneiss samples. In all samples quartz (0001) and (11-20) show
an asymmetry, which represents a sinistral motion indicating a top to the north sense of shear. This is in
accordance with literature and shows Zapport phase deformation in the Adula nappe (e.g. Löw, 1987;
Meyre et al., 1993; Pleuger et al., 2003). Different orientation patterns of quartz pole figures (10-11) and
(01-11) may be attributed to mechanical Dauphiné twinning, or induced by active rhombohedral slip (e.g.
Stipp and Kunze, 2008; Wenk et al., 2019). Both biotite and white mica show a strong CPO with a
pronounced alignment of their basal planes within the foliation in the gneiss samples (Fig. 4A). It should
be noted that in texture analysis (and in texture-based modeling of elastic properties) monoclinic crystals
are commonly defined in a first monoclinic setting (Matthies and Wenk, 2009), while a more common
second setting is used in this manuscript with (001) as a cleavage plane of mica. Both pagioclase and
kalifeldspar show a very weak to random CPO with only a few exceptions.
The marble sample yields a distinct calcite and dolomite CPO. Calcite exhibits an alignment of (0001) in Z-
direction and an alignment of (11-20) in X-direction (Fig. 4C). Both (0001) and (11-20) of dolomite show an
angle to the Z- and Y-direction respectively. In the metabasites, hornblende is the only mineral yielding a
pronounced CPO (Fig. 4D). It shows a strong alignment of (010) in Z-direction and (001) in X-direction in
both samples.

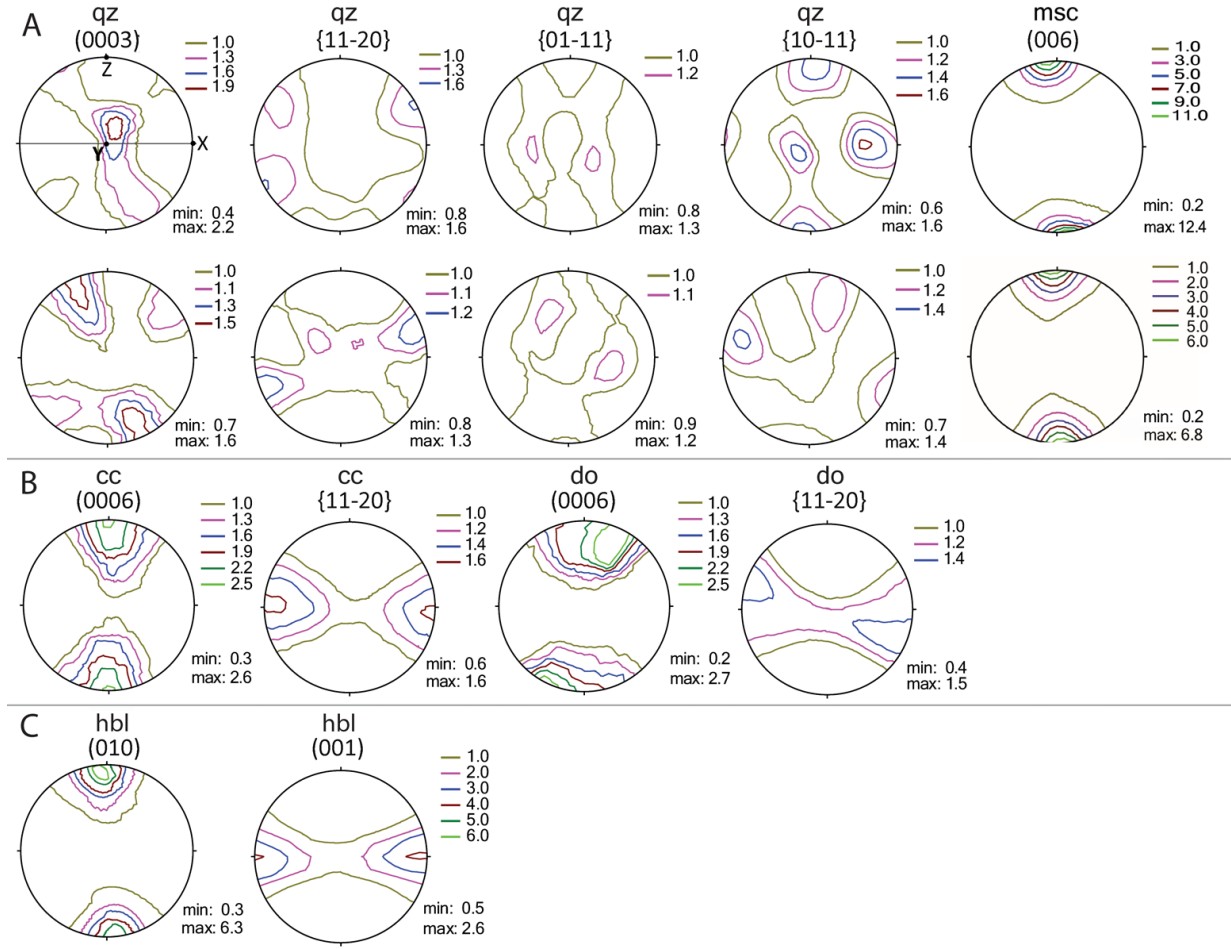

Figure 4: CPO types in the sample set (A) Common quartz (top: RK15-28; bottom: JK6) and mica (top: RK15-5; bottom: RK15-28) CPO in the orthogneisses and paragneisses; (B) calcite and dolomite CPO in the marble sample (RK68); (C) typical hornblende CPO in the metabasites (RK15-4). All pole figures are lower hemisphere equal area projections. The foliation normal (Z) is vertical, the lineation (X) is horizontal and north is left.

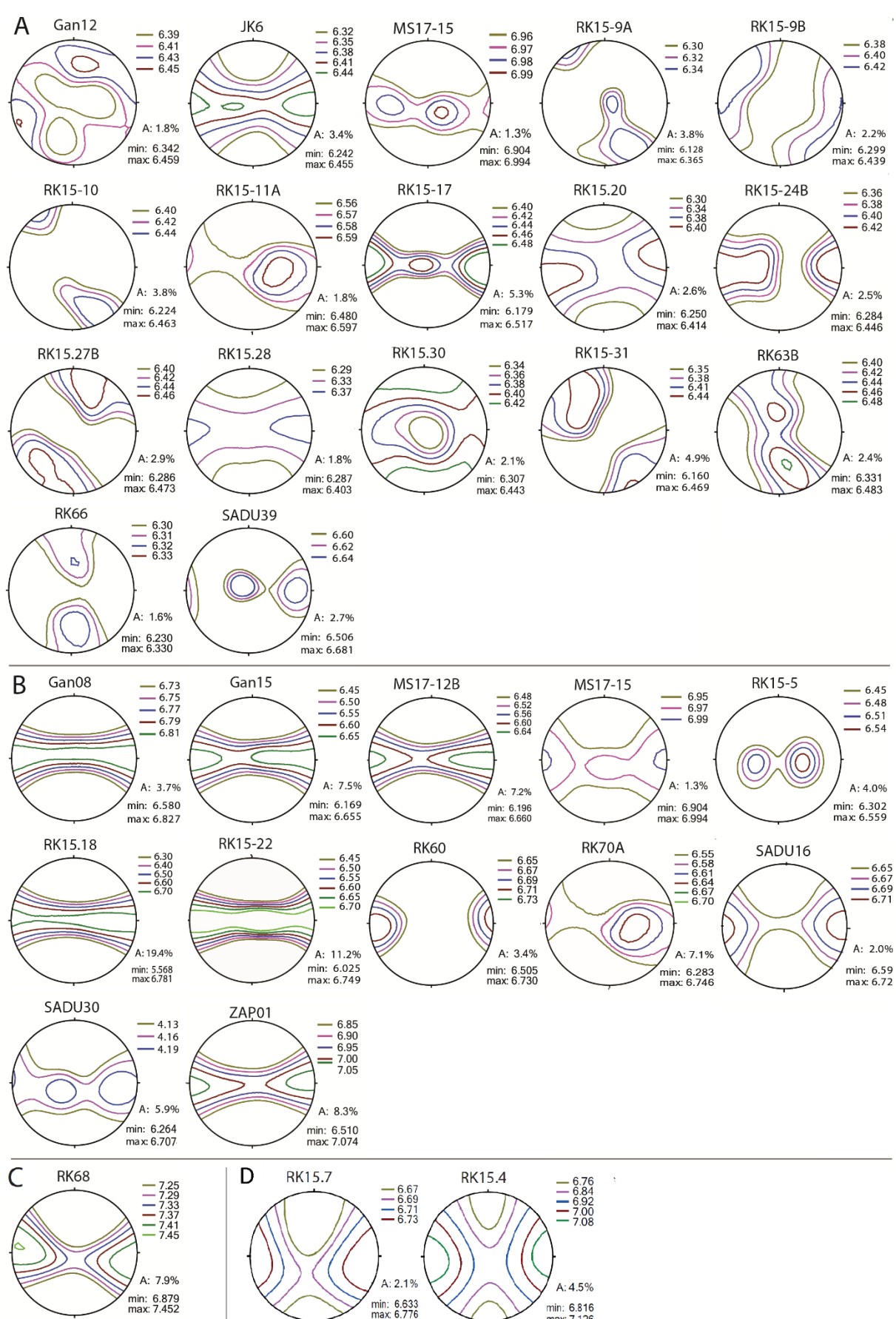

Figure 5: Modelled P-wave anisotropies of all natural samples in equal area stereographic projection. (A) Orthogneisses; (B) paragneisses ; (C) marble and (D) metabasites. Contour lines, as well as minima and maxima are in km/sec. The foliation is perpendicular to the projection plane, the lineation is horizontal. XYZ orientation is the same as in Fig. 4.

## 5.2. Modeled elastic anisotropies of natural samples

### 5.2.1. Orthogneisses

P-wave anisotropy ($AV_P$) is defined as A = ($V_P$max – $V_P$min)/ $V_P$mean *100%. The orthogneisses show two main patterns, one of which yields highest P-wave velocity ($V_P$) at an angle to the foliation normal, the other exhibits a $V_P$ maximum in the lineation direction with a distribution of high $V_P$ values in the foliation plane in some samples (Fig. 5A). The maxima at an angle to the foliation normal are frequently elongated or even show two distinct maxima within an area of higher $V_P$ (GAN12, RK15-9A, RK15-27B). Only few samples deviate from these two patterns showing maxima between the Y direction and the foliation normal (RK63B, RK66) or several maxima within the foliation plane (MS17-15, SADU39). $AV_P$ lies between 1.3 and 5.3% with an average of 2.9%. $V_P/V_S$ ratios are between 1.51 and 1.67 (Table 2A) with an average of 1.60.

| A | Vp A (%) | Vs1 A (%) | Vs2 A (%) | VP/Vs | Vp (km/s) | Vs (km/s) |
|---|---|---|---|---|---|---|
| GAN12 | 2,5 | 1,8 | 1,7 | 1,57 | 6,40 | 4,07 |
| JK6 | 3,4 | 2,0 | 3,0 | 1,63 | 6,35 | 3,91 |
| MS17-15 | 1,3 | 1,0 | 0,5 | 1,63 | 6,95 | 4,27 |
| RK15-10 | 3,8 | 3,1 | 2,4 | 1,53 | 6,35 | 4,16 |
| RK15-11A | 1,8 | 1,0 | 1,2 | 1,61 | 6,55 | 4,06 |
| RK15-17 | 5,3 | 5,0 | 2,5 | 1,65 | 6,35 | 3,86 |
| RK15-20 | 2,6 | 1,5 | 1,1 | 1,60 | 6,33 | 3,96 |
| RK15-24B | 2,5 | 1,8 | 2,1 | 1,64 | 6,37 | 3,89 |
| RK15-27B | 2,9 | 3,6 | 2,3 | 1,54 | 6,39 | 4,14 |
| RK15-28 | 1,8 | 1,1 | 1,6 | 1,64 | 6,35 | 3,86 |
| RK15-30B | 2,1 | 1,0 | 0,6 | 1,67 | 6,39 | 3,81 |
| RK15-31 | 4,9 | 5,4 | 3,6 | 1,51 | 6,33 | 4,18 |
| RK15-9A | 3,8 | 3,2 | 3,5 | 1,54 | 6,26 | 4,06 |
| RK15-9B | 2,2 | 1,5 | 1,5 | 1,57 | 6,38 | 4,07 |
| RK63B | 2,4 | 1,6 | 1,9 | 1,63 | 6,42 | 3,95 |
| RK66 | 1,6 | 1,1 | 0,8 | 1,64 | 6,29 | 3,83 |
| SADU39 | 3,1 | 2,7 | 2,0 | 1,56 | 6,57 | 4,21 |

| B | Vp A (%) | Vs1 A (%) | Vs2 A (%) | VP/Vs | Vp (km/s) | Vs (km/s) |
|---|---|---|---|---|---|---|
| GAN08 | 3,7 | 5,0 | 1,4 | 1,64 | 6,73 | 4,11 |
| GAN15 | 7,5 | 6,7 | 4,5 | 1,58 | 6,43 | 4,07 |
| MS17-12B | 7,2 | 5,7 | 3,3 | 1,57 | 6,44 | 4,10 |
| MS17-12C | 2,0 | 1,2 | 1,2 | 1,65 | 6,61 | 4,01 |
| RK15-18 | 20,5 | 19,4 | 11,5 | 1,65 | 6,18 | 3,73 |
| RK15-22 | 11,2 | 8,0 | 5,5 | 1,56 | 6,42 | 4,10 |
| RK15-5 | 4,3 | 4,0 | 1,9 | 1,55 | 6,42 | 4,14 |
| RK60 | 3,4 | 2,8 | 1,5 | 1,64 | 6,60 | 4,03 |
| RK68 | 7,9 | 4,4 | 2,3 | 1,82 | 7,22 | 3,96 |
| RK70A | 7,1 | 6,1 | 4,3 | 1,61 | 6,53 | 4,06 |
| SADU16 | 2,0 | 1,5 | 1,0 | 1,60 | 6,65 | 4,15 |
| SADU30 | 6,8 | 5,9 | 2,7 | 1,60 | 6,50 | 4,07 |
| ZAP01 | 8,3 | 6,6 | 3,3 | 1,64 | 6,81 | 4,17 |

| C | Vp A | Vs1 A | Vs2 A | VP/Vs | Vp | Vs |
|---|---|---|---|---|---|---|
| RK15-4B | 4,5 | 1,7 | 1,2 | 1,79 | 6,94 | 3,88 |
| RK15-7 | 2,1 | 0,6 | 0,7 | 1,76 | 6,70 | 3,81 |

Table 2: P-wave and S-wave anisotropy, VP/VS ratio as well as Voigt average of P-wave and S-wave velocities of (A) orthogneisses, (B) metasediments and (C) metabasites.

### 5.2.2. Paragneisses

The paragneiss samples all show highest $V_P$ value within the foliation plane (Fig. 5B). Most samples also yield a maximum in the lineation direction. There are two samples displaying maxima within the foliation plane but not aligned in the lineation direction (RK15-5; SADU30). The $AV_P$ of the paragneisses is highly variable ranging from 2.0% to 20.5% (Table 2B). Most samples, however, show a moderate $AV_P$ of 7-8%. $V_P/V_S$ ratios lie between 1.55 and 1.65.


### 5.2.3. Minor lithologies

The marble sample RK68 exhibits an $AV_P$ of 7.9% with a maximum at a small angle to the lineation direction
and some distribution of high $V_P$ values in the foliation plane (Fig. 5C). Its $V_P/V_S$ ratio is 1.82 (Table 2B).
The $V_P$ distributions in the metabasites show a pronounced maximum in the lineation direction (Fig. 5
D). Lowest $V_P$ is found parallel to the foliation normal. $AV_P$ values are 4.5% and 2% with $V_P/V_S$ ratios of 1.79
and 1.76, respectively (Table 2C).

### 5.3. Measured elastic anisotropies of natural samples

The $V_P$ distribution of the two gneiss samples, which were measured using ultrasound at different confining
pressures both show high $V_P$ in the foliation plane. The orthogneiss RK15-17 yields a maximum $V_P$ within
the foliation plane at a slight angle to the lineation (Fig. 6A). At maximum pressures of 400 MPa its $AV_P$ is
14%. The paragneiss RK15-22 was measured at a maximum pressure of 300 MPa. Maximum $V_P$ is aligned
in the lineation direction (Fig. 6B). It exhibits an $AV_P$ of 17%. Both samples show increasing $V_P$ values as
well as decreasing $AV_P$ coefficients with increasing pressures during the experiment (Table 3). In general,
the RK15-17 orthogneiss is elastically more isotropic and shows $V_P$ values comparable to the RK15-22
paragneiss at pressures over 100 MPa (Table 3), but at lower pressures P-wave velocities in the orthogneiss
decrease drastically, and the elastic anisotropy significantly increases, reaching values much higher than
in the paragneiss.

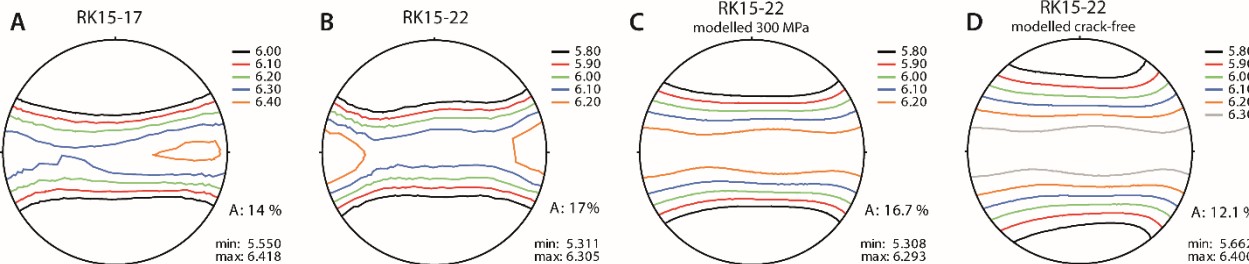

Figure 6: P-wave anisotropies of (A) an orthogneisses (RK15-17) and (B) a paragneiss (RK15-22) measured
using ultrasounding. Figures show P-wave distribution at maximum pressures in the experiments. $V_P$
distribution of RK15-22 modelled with GMS algorithm at 300MPa (C) and at crack-free pressures (D).
Contour lines, as well as minima and maxima are in km/sec. XYZ orientation is the same as in Figs. 4 and
410 5.


### 5.4. "Average" rock concept and crack-free "average" rock

Elastic properties and elastic wave velocities in rocks are normally assessed in laboratory measurements
on samples of several cm length. To implement elastic anisotropies in geophysical models these
laboratory-derived elastic properties need to be upscaled to a km scale. It is necessary to calculate elastic
properties of the rock massif in a long-wavelength approximation (Berryman, 1980), and thus a whole rock
massif may be represented as an effective "average" rock. It should feature average CPOs, volume
fractions and grain shapes of minerals, as well as average pore and crack patterns. Of course one needs to
bear in mind that even in these larger massifs heterogeneities like the aforementioned lenses and layers
of different lithologies exist.

| Pressure (MPa) | **A** RK15-17, experiment | | | **B** RK15-22, experiment | | | **C** RK15-22, model | | | Type I crack density | Type II crack density | Total crack porosity |
|---|---|---|---|---|---|---|---|---|---|---|---|---|
| | $V_{Pmin}$ (km/s) | $V_{Pmax}$ (km/s) | $AV_P$ (%) | $V_{Pmin}$ (km/s) | $V_{Pmax}$ (km/s) | $AV_P$ (%) | $V_{Pmin}$ (km/s) | $V_{Pmax}$ (km/s) | $AV_P$ (%) | | | |
| 0 | | | | 2.876 | 5.062 | 53 | | | | | | |
| 2 | | | | 3.292 | 5.134 | 43 | | | | | | |
| 10 | 2.637 | 4.459 | 51 | 3.904 | 5.349 | 31 | 3.904 | 5.350 | 31 | 0.205 | 0.056 | 0.0145 |
| 20 | 3.207 | 4.812 | 40 | 4.194 | 5.520 | 27 | 4.196 | 5.524 | 27 | 0.162 | 0.048 | 0.0118 |
| 50 | 4.106 | 5.316 | 26 | 4.584 | 5.786 | 23 | 4.583 | 5.791 | 23 | 0.112 | 0.031 | 0.0079 |
| 100 | 4.787 | 5.824 | 20 | 4.902 | 6.019 | 20 | 4.915 | 6.037 | 20 | 0.074 | 0.014 | 0.0046 |
| 200 | 5.307 | 6.203 | 16 | 5.202 | 6.269 | 19 | 5.213 | 6.263 | 18 | 0.043 | 0 | 0.0018 |
| 300 | 5.501 | 6.339 | 14 | 5.311 | 6.305 | 17 | 5.307 | 6.293 | 17 | 0.033 | 0 | 0.0014 |
| 400 | 5.550 | 6.418 | 14 | | | | | | | | | |
| Crack free | | | | | | | 5.662 | 6.400 | 12 | 0 | 0 | 0 |

Table 3: Results of ultrasonic measurements of (A) orthogneiss RK15-17 and (B) paragneiss RK15-22
showing $V_P$ and $AV_P$ at increasing pressures. (C) $V_P$ and $AV_P$ of RK15-22 modelled with GMS algorithm.
As a first approximation to the crustal properties, only major minerals were considered for the "average"
rock: plagioclase, muscovite and quartz. Minor or uncommon mineral phases were omitted. From the
selection of 30 natural crustal rocks, we identified characteristic CPO types and average CPO strengths for
all common mineral phases. In general, feldspar shows weak to random CPO, even in strongly deformed
samples. Furthermore, only minor differences have been observed between plagioclase and kalifeldspar.
Therefore, the ODF of a representative plagioclase with weak texture was chosen for the "average" rock,
namely, the albite ODF in RK15-28 sample. Since white mica is most common in both orthogneisses and
paragneisses, muscovite was chosen as representative mica for the average rock. In all samples mica shows
a pronounced alignment of its basal plane in the foliation. The mica ODF of two samples (RK15-5; RK15-
28) was combined in 1:1 ratio to yield an average preferred orientation for the "average" sample. Likewise,
the representative quartz ODF for the average sample was chosen as a combination of CPOs from two
different samples (JK6; RK15-28) in 5:6 ratio, based on the frequency of occurrence of each CPO pattern
in the sample set. These two samples show the typical quartz CPO patterns mentioned in section 4.4 and
shown in Fig. 4A.
Based on the analysis of all samples, average mineral volume percentages in gneisses (43% quartz, 40%
plagioclase, 17% mica) were considered for the "average" rock. Corresponding density value is 2670.7
kg/m$^3$. For the GMS method, grain shapes of minerals should be approximated by ellipsoids. Thin section
analysis of samples revealed more or less equiaxed grain shapes of quartz and feldspar and elongated mica
platelets with average aspect ratio of ≈0.2 (APPENDIX – Fig A1). Numerical models revealed that aspect
ratios of grains of mica and quartz within 0.1-1 range have only minor influence on bulk elastic properties
(Nishizawa and Yoshino, 2001; Vasin et al., 2013; Huang et al., 2021). Consequently, for the "average" rock,
we considered spherical grains of quartz and feldspar, and oblate spheroidal grains with axes ratio 1:1:0.2
for mica. As the shape of mica grains is related to cleavage, the corresponding SPO may be derived from
the CPO by considering additional rotation of the crystallite coordinate system (Vasin et al., 2013).
The preferred orientations, mineral volume fractions and grain shapes were combined in a model of the
elastic properties for the "average" rock using the GMS approach. The $V_P$ distribution in a crack-free
"average" rock is shown in Figure 7. There is a distribution of high $V_P$ values within the foliation plane, and
the maximum $V_P$ direction is located between the lineation (X-direction) and the Y-direction. The $AV_P$ of
the "average" crack-free gneiss is 4%.
This "average" rock would be found at depths of at least 28 km, which means that considering an average
crustal thickness most of the crust would be above this point. This is why it is also important to consider
the microcrack pattern in such an average rock at lower depth.

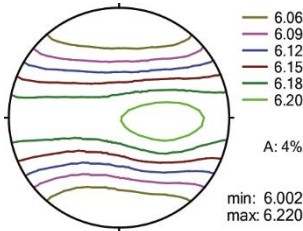

Figure 7: Modelled P-wave anisotropies of an average gneiss at 740 MPa. Contour lines, as well as minima
and maxima are in km/sec. XYZ orientation is the same as in Fig. 4.
5.5. "Average" rock with microcrack systems
As directly evident from thin sections analysis (Figure 3), low aspect ratio microcracks are present in the
samples. At low overburden depths, these microcracks are open. As seen from Table 3, at low pressures
measured elastic wave velocities are decreased and elastic anisotropy is increased compared to the high
pressure, where the majority of microcracks is closed. To account for the change in elastic anisotropy of
the "average" rock due to pressure/depth changes, it is necessary to include these microcracks and their
closure with increasing pressure into the model.

| Pressure (MPa) | Depth (km) | Density (kg/m³) | Type I crack density | Type II crack density | Total crack porosity | $V_{Pmin}$ (km/s) | $V_{Pmax}$ (km/s) | $AV_P$ (%) |
|---|---|---|---|---|---|---|---|---|
| 5 | 0.2 | 2627.19 | 0.246 | 0.057 | 0.0163 | 4.203 | 4.728 | 12 |
| 10 | 0.4 | 2631.99 | 0.205 | 0.056 | 0.0145 | 4.442 | 4.891 | 10 |
| 20 | 0.8 | 2639.21 | 0.162 | 0.048 | 0.0118 | 4.721 | 5.121 | 8 |
| 50 | 1.9 | 2649.62 | 0.112 | 0.031 | 0.0079 | 5.077 | 5.454 | 7 |
| 100 | 3.8 | 2658.43 | 0.074 | 0.014 | 0.0046 | 5.372 | 5.748 | 7 |
| 200 | 7.6 | 2665.91 | 0.043 | 0 | 0.0018 | 5.630 | 6.012 | 7 |
| 300 | 11.5 | 2666.98 | 0.033 | 0 | 0.0014 | 5.709 | 6.057 | 6 |
| ≈740 | 28.2 | 2670.72 | 0 | 0 | 0 | 6.002 | 6.220 | 4 |

Table 4: $V_P$, $AV_P$, and crack densities of "average" rock model at increasing pressures and corresponding
depth. 740 MPa is an estimation of the pressure where the cracks are closed (see text). Cracks type I have
the same ODF as mica.
As a first approximation, we considered that the "average" rock should have the same crack distribution
as one of the characteristic gneiss samples, i.e., sample RK15-22. From thin section analysis (Figure 3D-F),
two possible microcrack systems were identified. There is one set of microcracks mostly oriented along
the muscovite platelets, and we denote this set as type I cracks. Type I cracks were assumed to have the
same SPO as muscovite grains; and their aspect ratio was estimated to be ≈0.01 (Appendix – Fig A2). As
these cracks are roughly parallel to mica platelets, within the GMS algorithm type I cracks were
approximated by oblate ellipsoids with an axial ratio of 1:1:0.01. Another set of cracks – denoted as type
II cracks - intersects quartz grains. These cracks are mostly oriented parallel to the Z axis. They display a
broader range of aspect ratios with an average of ≈0.025 (APPENDIX – Fig A1). Since these cracks are
mostly within equiaxed quartz grains, they were approximated by oblate ellipsoids with an axial ratio of
1:1:0.025. To determine the changes of crack densities of type I and II with pressure, the following
procedure was applied.
Elastic properties of a crack-free RK15-22 gneiss were modelled with GMS algorithm using measured ODFs,
mineral volume fractions (55% quartz, 15% albite and 30% muscovite), and assuming spherical grain
shapes for quartz and albite, and 1:1:0.2 ellipsoidal grains for muscovite. Using mineral density values from
the same references as mineral single crystal elastic properties, a density of 2702.3 kg/m$^3$ was computed
for a crack-free RK15-22. Using model elastic properties and this density value, the $V_P$ distribution in a
crack-free RK15-22 was calculated (Fig. 6C).
Density measurements of RK15-22 at atmospheric pressure yield a value of 2658 kg/m$^3$. Thus, crack
porosity in RK15-22 is restricted to a maximum of about ~1.7%. Consequently, type I and type II cracks
were added into the model crack-free RK15-22 gneiss to reproduce measured $V_P$ distributions at different
pressures, similar to the procedure of porous polycrystalline graphite (Matthies, 2012). The only varying
parameters are the type I and type II crack porosities, with the total crack porosity within the
aforementioned limit. Using this procedure, an adequate description of experimental $V_P$ distributions with
the GMS approach was achieved at all pressures above 2 MPa. The wave velocities and $AV_P$ values of RK15-
22 models are given in Table 3.

| Pressure (MPa) | $C_{11}$ (GPa) | $C_{12}$ (GPa) | $C_{13}$ (GPa) | $C_{14}$ (GPa) | $C_{15}$ (GPa) | $C_{16}$ (GPa) | $C_{22}$ (GPa) | $C_{23}$ (GPa) | $C_{24}$ (GPa) | $C_{25}$ (GPa) | $C_{26}$ (GPa) | $C_{33}$ (GPa) | $C_{34}$ (GPa) | $C_{35}$ (GPa) | $C_{36}$ (GPa) |
|---|---|---|---|---|---|---|---|---|---|---|---|---|---|---|---|
| 5 | 58.0 | 9.1 | 7.8 | 0.1 | -0.1 | -0.2 | 58.1 | 7.9 | 0.2 | -0.1 | -0.3 | 46.4 | 0.1 | -0.1 | 0.0 |
| 10 | 62.2 | 10.2 | 9.0 | 0.1 | -0.1 | -0.2 | 62.4 | 9.0 | 0.2 | -0.1 | -0.3 | 51.9 | 0.1 | -0.1 | 0.0 |
| 20 | 68.4 | 11.8 | 10.6 | 0.1 | -0.1 | -0.2 | 68.6 | 10.6 | 0.2 | -0.1 | -0.3 | 58.8 | 0.1 | -0.1 | 0.0 |
| 50 | 77.9 | 14.6 | 13.3 | 0.1 | 0.0 | -0.2 | 78.2 | 13.3 | 0.2 | -0.1 | -0.3 | 68.3 | 0.1 | -0.1 | 0.0 |
| 100 | 86.8 | 17.5 | 16.1 | 0.1 | 0.0 | -0.1 | 87.1 | 16.2 | 0.2 | -0.1 | -0.4 | 76.7 | 0.1 | -0.1 | 0.0 |
| 200 | 95.2 | 20.7 | 19.1 | 0.1 | 0.0 | -0.1 | 95.6 | 19.1 | 0.2 | -0.1 | -0.4 | 84.5 | 0.0 | -0.1 | 0.0 |
| 300 | 96.7 | 21.3 | 19.8 | 0.1 | 0.0 | -0.1 | 97.1 | 19.9 | 0.2 | -0.1 | -0.4 | 86.9 | 0.0 | -0.1 | 0.0 |
| ≈740 | 102.2 | 23.7 | 22.9 | 0.0 | 0.1 | -0.1 | 102.6 | 23.0 | 0.1 | -0.1 | -0.4 | 96.2 | 0.0 | -0.1 | 0.0 |

continuation

| Pressure (MPa) | $C_{44}$ (GPa) | $C_{45}$ (GPa) | $C_{46}$ (GPa) | $C_{55}$ (GPa) | $C_{56}$ (GPa) | $C_{66}$ (GPa) |
|---|---|---|---|---|---|---|
| 5 | 21.5 | -0.1 | -0.1 | 21.4 | 0.1 | 24.6 |
| 10 | 23.4 | -0.1 | -0.1 | 23.2 | 0.1 | 26.2 |
| 20 | 25.8 | -0.1 | -0.1 | 25.6 | 0.1 | 28.5 |
| 50 | 29.1 | -0.1 | -0.1 | 28.9 | 0.1 | 31.9 |
| 100 | 32.0 | -0.1 | -0.1 | 31.7 | 0.1 | 35.0 |
| 200 | 34.5 | -0.1 | -0.1 | 34.2 | 0.1 | 37.7 |
| 300 | 35.1 | -0.1 | -0.1 | 34.8 | 0.1 | 38.1 |
| ≈740 | 37.3 | -0.1 | -0.1 | 36.9 | 0.1 | 39.7 |


Table 5: Bulk elastic tensor components of the "average" rock model, rounded to first decimal digit.

At maximum pressure of 300 MPa the experimental $V_P$ values are 0.3-0.7 km/s lower than corresponding
velocities in the crack-free RK15-22 with biggest differences for minimum velocities. This implies a small
amount of open microcracks in the experiment. Modeling suggests that type II cracks with 0.025 aspect
ratio are not necessary to describe bulk elastic properties of RK15-22 sample at pressures of 200 MPa and
higher. Thus, we assume that type II crack porosity is close to zero at 300 MPa. Since type I cracks
orientation distribution is not random, and the material is elastically anisotropic with $AV_P$ =17%, only a
rough estimation of type I cracks closure pressure can be made. We averaged the stiffness tensor of crack-
free RK15-22 over all directions and applied the relation derived by Walsh (1965) for an isotropic rock to
obtain a closure pressure of ≈740 MPa for type I cracks at an aspect ratio of 0.01.
It is recognized that at low crack porosities effective elastic properties of the material depend on the crack
density, while crack porosity is irrelevant (Vernik, 2016; Kachanov and Mishakin, 2019). Crack porosity and
crack density may be related for certain types and distributions of cracks. E.g., in the case where all cracks
have the same aspect ratio, as type I or type II pores separately, there is a simple equation (Lokajicek et
al., 2021) connecting crack porosity and crack density. Thus, in Table 4, crack densities are given for type I
and type II cracks separately, as well as the total crack porosity. We assume that the same system of cracks
exists in an "average" sample such as RK15-22, with the same orientation distribution and the same crack
density values at corresponding confining pressure. The GMS algorithm was used to add this crack system
to the crack-free "average" rock, and the density of the crack-free "average" rock was used to estimate
the overburden from the pressure values. From that, the dependencies of all stiffness tensor components
of the "average" rock on depth were obtained, as well as the elastic wave velocities and the $AV_P$
coefficients (Tables 4 and 5).
We note that the proposed model is aimed to reproduce ultrasonic wave velocities measured during
sample loading. It may be expected that during unloading, ultrasonic wave velocities would be higher at
same pressure levels due to irreversible closure of some microcracks. This effect would certainly adjust
the depth estimates, but it may also change the rock anisotropy if the mechanism of irreversible closure is
different for type I and type II cracks. The effect of crack closure should be studied in more detail with
respect to rock massif.

6. Discussion

There are various factors influencing the elastic anisotropy of rocks. While the deformation-induced CPO
is the main cause, there are other aspects like shape preferred orientation (SPO) of grains, or layering
contributing to elastic anisotropy. Another important factor influencing elastic anisotropy, especially at
lower depth is the occurrence of microcracks. In the following, we discuss the elastic anisotropies -
calculated and measured - of the natural samples from this study. We will elaborate the applicability of
the model "average" rock to larger scale crustal rock units and critically assess the controlling factors of
the elastic anisotropy of crustal rocks.

6.1. Elastic anisotropy of natural samples

6.1.1. Orthogneisses

The AV$_P$ calculated from the CPO data of orthogneisses is largely influenced by CPOs of quartz and mica.
Since feldspar generally shows weak or no CPO, its presence in the samples mainly contributes to a
decrease in AV$_P$. Mica adds to increased V$_P$ values within the foliation plane as well as the maxima in the
lineation direction in some samples. Highest V$_P$ is found within the basal plane of mica single crystals,
which defines the V$_P$ pattern caused by observed alignment of mica basal planes within the foliation. The
maxima in the lineation direction are caused by a slight tilting of mica basal planes around the lineation.
This leads to broadening of high V$_P$ region within the YZ-plane and results in the highest V$_P$ in lineation
direction. Highest V$_P$ values of quartz single crystals are observed close to normals to their rhombohedral
planes. Patterns showing elongated V$_P$ maxima close to the periphery at an angle to the foliation and the
patterns with several maxima for V$_P$ are due to the influence of quartz CPO. The frequently observed
asymmetry in these patterns with respect to the reference frame of foliation and lineation reflects non-
coaxial deformation of the rocks. All units in the central Adula Nappe show a top-to-the-north sense of
shear (e.g. Nagel, 2008), thereby producing asymmetric quartz CPO, which in turn leads to the asymmetric
V$_P$ distributions in the mica-poor orthogneisses. Both AV$_P$ as well as V$_P$ patterns are similar to those in
previous studies, which either show high V$_P$ in the foliation with a maximum in the lineation direction
(Ivankina et al. 2005; Ullemeyer et al., 2006; Kern et al. 2008; Zel et al., 2015; Ivankina et al., 2017;
Schmidtke et al., 2021), at an angle to the lineation (Vasin et al., 2017), or elongated asymmetric maxima
between the foliation normal and the foliation plane (Ullemeyer st al., 2006; Llana-Fúnez et al., 2009).
The orthogneiss sample RK15-17 measured in the lab shows high V$_P$ distributed within the foliation plane
with a maximum at a slight angle to the lineation direction. While both the measured and the calculated
velocity patterns for this sample show high V$_P$ distributed in the foliation plane, the AV$_P$ pattern calculated
from CPO yields its maximum aligned in the lineation direction with an additional maximum in Y-direction.
The AV$_P$ coefficient calculated from measured P-wave velocities at a pressure of 400 MPa is higher than
the calculated one by a factor 2.6, which is mostly due to still open microcracks, not considered within the
Voigt averaging scheme. Due to a preferred orientation of microcracks parallel to the mica basal plane
(Fig. 3A-C) and an alignment of mica in the foliation V$_P$ is slower normal to the foliation and AV$_P$ is higher
in the samples measured in the lab, even at the highest pressures.
V$_P$/V$_S$ ratios in the orthogneiss samples are influenced by the volume percentage of the constituent mineral
phases. Due to the low Poisson ratio of quartz and its generally large volume percentage in the
orthogneisses their V$_P$/V$_S$ ratios of 1.51-1.67 are low.
6.1.2. Paragneisses
Like in the orthogneisses, the V$_P$ pattern of the paragneisses and mica schists is influenced by mica and
quartz CPO with a larger mica contribution due to its generally higher volume content in paragneisses
compared to the orthogneisses (Table 1B). Likewise, mica CPO leads to high V$_P$ values within the foliation
plane and frequently to a V$_P$ maximum in the lineation direction. This V$_P$ pattern is similar to that of
paragneisses in previous studies (e.g. Weiss et al., 1999; Erdman et al., 2013; Keppler et al., 2015;
Ullemeyer et al., 2018). V$_P$ patterns showing maxima within the foliation plane, but not aligned with the
lineation, are likely caused by a discrepancy between CPO formation of quartz and CPO formation of mica.
The samples are oriented according to their visible mineral stretching lineation, which was formed by
quartz in most samples. The alignment of high velocities is, however, caused mostly by mica CPO and
undulating mica grains around the stretching lineation.
The sample measured in the lab, RK15-22, similar to the case of the orthogneiss sample, shows a higher
influence of mica on $AV_P$ due to its alignment in the foliation and similarly oriented microcracks. While in
the calculated $V_P$ distribution, high velocities are distributed within the foliation plane, the measured
velocities show a distinct maximum in the lineation direction. The measured version also shows a higher
$AV_P$ than the one calculated from the CPO. The difference, however, is not as large as for the orthogneiss
sample. In case of the paragneiss sample the measured $AV_P$ is higher than the calculated one by a factor
of 1.5. Similar to the orthogneisses, this value is well in the range of published data comparing
experimental and modeled anisotropy. While experimental anisotropies are always higher than the ones
modeled using CPO, the factor is variable for gneiss samples ranging from 1.3 (e.g. Vasin et al., 2017) to
6.6 (e.g. Ullemeyer et al., 2006). Considering experimental and modeled elastic anisotropy data of 18
gneiss samples from different studies, experimental anisotropy is 3 times higher than the modeled ones
on average (Ivankina et al., 2005; 2017; Punturo et al., 2005; Ullemeyer et al., 2006; 2018; Kern et al., 2008;
Kern, 2009; Llana-Fúnez et al., 2009; Lokajicek et al., 2014; Zel et al., 2015; Vasin et al., 2017). $V_P/V_S$ ratios
of the paragneisses are determined by the volume percentage of quartz and yield values of 1.55-1.64.
Higher volume percentages of quartz lead to lower $V_P/V_S$ ratios.
Comparing the $V_P$ velocities calculated from the Voigt model (Figure 5B) and the GMS crack free model
(Table 3) of the RK15-22 sample, it is evident that the Voigt model velocities are ≈300-400 m/s higher. Yet,
symmetries of velocity distributions and $AV_P$ coefficients computed using these two models are quite close,
suggesting that the Voigt modeling is reliable to assess the degree of elastic anisotropy of gneisses.
Tables 3 and 4 demonstrate a correlation of measured ultrasonic wave velocities and their anisotropy in
RK15-22 gneiss as well as the GMS model based on the two types of cracks at pressures of 5-300 MPa as
presented before. At 2 MPa, and also at atmospheric pressure, the proposed model was not able to
correctly reproduce experimental $V_P$ patterns. At low confining pressure it is observed that both self-
consistent and non-interactive theories may be inadequate to describe the elastic velocity behavior, which
might be due unknown crack geometries (Hadley, 1976). It is likely that another system of thinner
microcracks is required to match the GMS model and experimental ultrasonic wave velocities in RK15-22
at very low confining pressure.
As expected, the GMS models of RK15-22 at higher pressure require lower crack densities/porosities to
describe the experimental ultrasonic data. Modeling suggests that thinner type I cracks are closed at a
faster rate with increasing pressure compared to thicker type II cracks. Yet, due to an initially lower crack
density of type II cracks, the modeling suggests that their influence on the bulk elastic properties of model
RK15-22 gneiss becomes negligible at and above a pressure of 200 MPa. In contrast, type I cracks are
necessary to match the experimental and model P-wave velocities at a pressure of 300 MPa. To estimate
the closing pressure of type I cracks, we disregarded RK15-22 elastic anisotropy and calculated average
Young's modulus and Poisson ratio of the gneiss. According to the simple model of crack closure in the
isotropic rock (Walsh, 1965), the closing pressure of type I cracks is ≈740 MPa.
Naturally, the proposed model based on laboratory measurements of rock properties is quite simplistic,
with some limitation coming from the modelling method itself, and others related to available
experimental data. The GMS treats material as an infinite effective medium, which is filled by ellipsoidal
inclusions without gaps or overlaps. Local heterogeneities, stress concentrators arising, e.g., on grain
boundaries, correlations in grain positions or orientations, or size-related effects are not considered. For
the "average" rock, accessory phases were discarded, and the most characteristic ODFs, volume fractions
and grain shapes of main minerals were used assuming that the studied set of samples represents the
Adula Nappe sufficiently well. We assumed that microcrack systems and their closure with pressure in the
"average" rock is the same as in the paragneiss sample. A shape related distribution of microcracks,
deviations of the assumed SPOs of the cracks from those actually present in the gneiss, possible
dependence of microcrack SPO on shape of cracks, and changes of all these parameters with pressure,
including irreversible closure of different microcracks, are neglected. Our results suggest that even small
open crack densities at relatively high confining pressures have a notable influence on the elastic
anisotropy of the paragneiss. Therefore, a comprehensive and precise quantification of the microcrack
characteristics is necessary to simulate realistic models of pressure dependencies on the bulk elastic
properties of rocks.
6.1.3. Marble
In the marble sample, the maximum $V_P$ is at a small angle to the lineation caused by the influence of both
the dolomite and the calcite CPO. The $AV_P$ of marble in the literature is highly variable depending on the
grade of deformation (Burlini and Kunze, 1999; Zappone et al., 2000; Punturo et al., 2005; Schmidkte et
al., 2021). Since the marble lenses in the Adula Nappe only make up a few meters in thickness they do not
contribute to the overall elastic anisotropy of the unit to large amounts. Depending on the thickness and
distribution of such lenses or layers, they could be considered for carbonate-rich crustal models. The
sample yields a high $V_P/V_S$ ratio of 1.82, which is influenced by both calcite and dolomite. These high $V_P/V_S$
ratios are typical for marble (e.g. Keppler et al., 2015). The combination of high $V_P/V_S$ ratio, as well as high
$AV_P$ may constrain a very specific signal for marble-rich crust at depth and help to detect specific features
such as large subducted carbonate platforms.
6.1.4. Metabasites
The $AV_P$ of the metabasites is dominated by hornblende, which has the highest volume percentage and is
the only mineral showing a strong CPO. Highest VP is found within the lineation and caused by the
alignment of (001), which is close to the highest VP in hornblende single crystals. Due to the stronger
hornblende CPO of RK15-4, the $AV_P$ is higher in this sample. Studies on elastic anisotropies of metabasites
mainly focus on eclogites and blueschists (e.g. Abalos et al., 2011; Bezacier et al., 2010; Keppler et al.,
2017; Zertani et al., 2019). Many of the metabasic units exhumed during continental collision, however,
are strongly retrogressed with large amounts of amphibole and/or chlorite. Recent studies show that these
retrogressed rocks frequently show higher elastic anisotropy than pristine basalts, gabbros or also
eclogites due to higher elastic anisotropy of amphibioles compared to pyroxenes, as well as a pronounced
deformation during exhumation (e.g. Neufeld et al., 2008; Keppler et al., 2016; Park and Jung, 2020;
Schmidtke et al., 2021). $V_P/V_S$ ratios of 1.79 and 1.76 for RK15-4 and RK15-7, respectively, are typical for
metabasites (e.g. Worthington et al., 2013; Schmidtke et al., 2021).
6.2. Elastic anisotropy of the modeled "average" rock

Realistic upscaling of the rock elastic properties measured within limited scale or on laboratory samples
to the seismic scale is of a long-standing interest, e.g., in hydrocarbon reservoirs (Sayers, 1998; Bayuk et
al., 2008; Avseth et al., 2010). Here, we consider a rather homogeneous crystalline rock with low crack
porosity, and we try to build an effective large-scale model using features of the studied rock massif:
average mineral volume fractions, preferred orientations, grain shapes and microcracks systems.
As expected, the "average" rock shows a distribution of high $V_P$ values normal to the Z-axis due to the
preferred orientation of mica, with a maximum $V_P$ value at an angle to the X-axis due to the influence of
the preferred orientation of quartz (Fig. 7).  This is a common pattern in the natural sample set (Fig. 5A
and B). Some orthogneisses in the natural sample set show maxima at an angle to the foliation normal,
which is different from the average sample (Fig. 5A). However, these samples generally show a low $AV_P$
and do not strongly contribute to the overall anisotropy. The model suggests decreasing $AV_P$ and increasing
$V_P$ values with increasing depth due to the closure of microcracks. A crack free "average" rock has $V_P$ values
slightly over 6 km/s and a rather low $AV_P$ of 4%, which is in between $AV_P$ values characteristic for
paragneisses and orthogneisses. At lower confining pressure down to 5 MPa (corresponding to a depth of
≈ 200 m), the model suggests a decrease of $V_P$ values to ~4.5 km/s, and an increase of $AV_P$ to 12% (Table
4) due to open microcracks.

One of the main improvements of our model  is the better quantification of microcrack systems, as
explained in section 6.1.2. Crack closure with increasing pressure in anisotropic gneisses should be studied
in more detail to reliably expand the crack closure in RK15-22 paragneiss to large rock units in general. In
addition to crack closure due to pressure, microcracks in quartz grains may be sealed by solution-
precipitation processes (e.g. Brantley et al., 1990; Vollbrecht et al., 1991; Derez et al., 2015).
Microfractures parallel to the r- and z rhombohedral planes of quartz can heal after little or no shear
displacement (e.g. Menegon et al., 2008). These healed cracks frequently occur as fluid inclusions trails in
quartz grains. Experimentally deformed quartz showed that the trails are commonly arranged in planes
parallel to the compression axis (Stünitz et al., 2017). Some inclusion trails found in the current samples
could be part of the same process (APPENDIX - Fig. A2). Intragranular microcracks can also be
crystallographically controlled (Vollbrecht et al., 1999). Hence, when the CPO of quartz is strong, a
preferred microcrack alignment can also be related to certain crystallographic orientations.
Further model improvement may be achieved by more detailed constraints on the mineral volume
fractions and crystallographic textures within the rock massif via more extensive sampling.
The calculated "average" rock model is related to the XYZ coordinate frame, defined by rock foliation and
lineation. To improve the model, it is necessary to account for possible foliation or lineation direction
changes through the rock unit by relating all crystal and shape preferred orientations to the same global
reference frame, e.g., geographical coordinates.
It is evident that the calculated model of the "average" rock does not consider large scale layering. It may
be introduced into the model by creating "average" rock layers consisting of the characteristic minerals
with their preferred orientations and microstructures and using a Backus averaging to combine thin
(relative to the lateral size) rock layers into a seismic scale effective medium (Backus, 1962; Sayers, 1998).
Furthermore, large scale faults are an important factor when considering elastic anisotropy and have to
be considered in any model of the Alps. Finally, only confining pressure and the density of the crack-free
"average" rock were used to estimate the depth values. Compositional variations would change the depth
estimates.
Despite all these simplifications of the current model, in principle, the proposed "average" rock may be
constructed to represent effective elastic properties on any necessary scale if there is sufficient
information on modal composition, textures and microstructures available from the selected samples.
Then direct comparison of the "average" rock with seismic data on the uppermost layer of the crust can
be made.
6.3. Elastic anisotropy in the Alps
The well-studied geology of the Alps provides comprehensive foliation and lineation maps (e.g., Steck
1990). Surface data can be correlated with seismic imaging making it possible to construct models for
different tectonic structures at depth (e.g. Yosefnejad et al., 2017). For the present study, the Adula Nappe
was chosen as a representative unit for deformed crust in the Alps. The central part of the Adula Nappe,
where the samples for this study have been collected exhibits a shallowly NE dipping foliation and a NS
trending lineation mainly formed during peak pressure and early stages of exhumation. The northern and
southern parts of the nappe, however, have been overprinted by younger deformation (e.g. Löw, 1987;
Nagel, 2008; Kossak et al. 2017).
The afore mentioned discrepancy between quartz lineation and mica CPO in several of the samples has
not been well studied with respect to the seismic anisotropy. It could be a common issue for most upper
crustal units in the Alps, exhibiting a complicated deformation history. Hence, maxima for elastic
anisotropies in the lineation direction in mica rich rocks cannot simply be correlated to measurements in
the field.
Microcrack distribution and orientation have not been investigated systematically throughout the rock
units of the Alps and they might exhibit strong local variations corresponding to the large-scale fracture
and fault pattern (e.g. Vilhelm et al., 2010). This has also a great effect on the travel times of P- and S-
waves, i.e. $V_P$ and $V_S$ are significantly decreased (e.g. Yan et al., 2005; Kelly et al., 2017) and therefore
needs to be considered for any large-scale section or model of the Alps (e.g. TRANSALP: Lüschen et al.,
2004; Millahn et al., 2005; AlpArray: Hetényi et al., 2018; Molinari et al., 2020).
While deformed granitoids (e.g. orthogneisses) and deformed clastic metasediments (e.g. paragneisses)
are the dominant lithologies, the rock spectrum found in the Alps and other collisional orogens ranges
from sedimentary rocks as well as metasediments like marbles, micaschists and quartzites, over
metabasites like eclogites, blueschists, amphibolites and greenschists to ultrabasic rocks like peridotites
and serpentinites. These lithologies might occur as small layers within the larger gneiss massifs
contributing to the overall seismic properties, but they also occur throughout the Alps as large coherent
units, which have to be considered. Furthermore, volcanic and plutonic intrusions are a common
occurrence in collisional orogens.
There are several nappes within the Alps dominated by (meta-)basic (e.g. the Zermatt–Saas zone:
Angiboust et al., 2009) and ultrabasic rocks (e.g. the Ivrea Complex: Hartmann and Wedepohl, 1993), which
have to be considered in some seismic profiles across the Alps. While we present data on some common
minor lithologies, like amphibolite, marble and micaschist, we also refer the reader to data on
metasediments (e.g. Punturo et al., 2005), metabasites (e.g. Abalos et al., 2011; Bezacier et al., 2010;
Zertani et al., 2020; Schmidtke et al., 2021) and ultrabasic rocks (Mainprice et al. 2000; Ullemeyer et al.,
2010). Within the NFP20 EAST profile considered in the present study, amphibolites and marbles mostly
occur as small lenses of under 1 km of thickness. Detecting them within the bulk of paragneisses and
orthogneisses is less likely. If they produce a seismic anisotropy signal will depend, of course, on the
seismic wave length. Zertani et al. (2020), for example, used the finite element method to employ eclogite
facies shear zones within granulites in models for petrophysical properties. In the present work, however,
we consider the elastic anisotropy of major gneiss units most critical for the investigated part of the section
and other rock units are negligible because of their small volume proportion. Therefore smaller lithological
variations as well as geometrical irregularities have been ignored for the overall model. Gneiss samples in
this as well as previous studies generally show an alignment of high $V_P$ within the foliation plane. That is
why the foliation of gneisses and mica schists formed during continental collision and exhumation is likely
a main factor controlling the elastic anisotropies of the continental crust in collisional orogens. The data
presented in this study yield a first approximation for average crustal seismic properties with increasing
depth as well as the specific seismic property spectrum of this deformed upper crustal section of the Alps.
7. Summary and Conclusion
1. The investigation of a large set of rocks collected in the Adula Nappe, which is considered to be
representative of deformed upper crustal rocks in the Alps, indicates a large variety of elastic anisotropies.
2. The Adula Nappe is mostly made up of orthogneisses with modelled AVP between 1.3 and 5.3% and
Vp/Vs ratios between 1.51 and 1.67, as well as paragneisses with modelled AVP between 2.0% to 20.5%
and Vp/Vs ratios between 1.55 and 1.64.
3. Metabasites that make up only 100 m thick lenses, show an AVP of 2-4.5% and VP/VS ratios of  1.76-
1.79. Marble lenses of even smaller dimensions yield an AVP of 3.4% and VP/VS ratio of 1.83. Yet, these
lenses are statistically of small significance for the considered section of para- and orthogneisses.
4. Orthogneiss and paragneiss measured in the lab using ultrasound both show higher AVP as well as lower
VP compared to the ones modelled using CPO, which is caused by open microcracks in the rocks at shallow
depth.
5. Average elastic anisotropies were calculated for a typical gneiss using common CPO types of constituent
mineral phases, mineral content, grain shapes and crack systems within the sample set. Calculated elastic
constants are considered to be representative for the range of depths from a few hundred meters up to
≈28 km. The modelled "average" gneiss yields an $AV_P$ of 4% at a depth of ≈28 km, where the vast majority
of microcracks is closed. Due to the opening of microcracks, the elastic anisotropy of the model gneiss
increases towards shallower depth and reaches $AV_P$ = 12% at ≈0.2 km. This makes it possible to either
choose parameters of an average sample representative of rocks at depths higher than 28 km, or choose
an average sample at increasingly lower depth with progressively opening microcracks, depending on the
depth of interest.


Acknowledgement

We are grateful for the very constructive and elaborate reviews by Benito Abalos, Sascha Zertani, as well
as the anonymous reviewer. These reviews strongly improved our manuscript. This study was funded by
the German research foundation (DFG-grant No. KE 2268/2-1, STI298/9-1) as part of the DFG priority
programme "Mountain Building Processes in 4 Dimensions". Fruitful discussions within the priority
programme are gratefully acknowledged. Furthermore, this study was partially supported by the Czech
Science Foundation research grants 18-08826S, 21-26542S and by the Czech Academy of Sciences project
RVO 67985831. Authors appreciate the access to the SKAT diffractometer at FLNP JINR. The project was
partially supported by the JINR theme No. 04-4-1121-2015/2020.

 APPENDIX

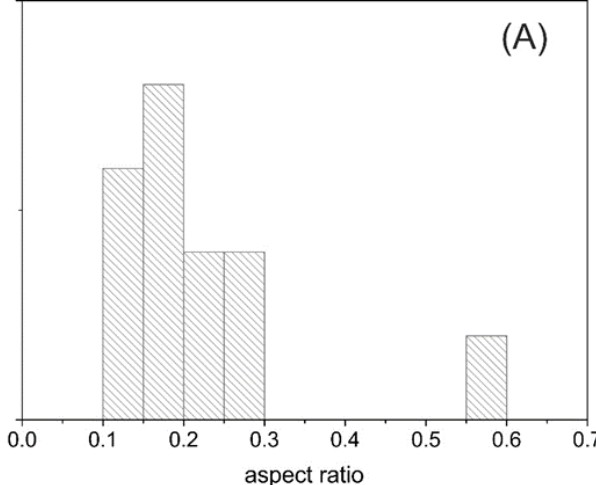

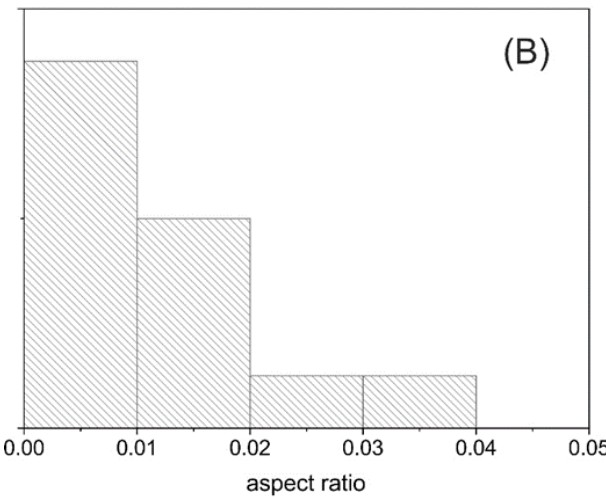

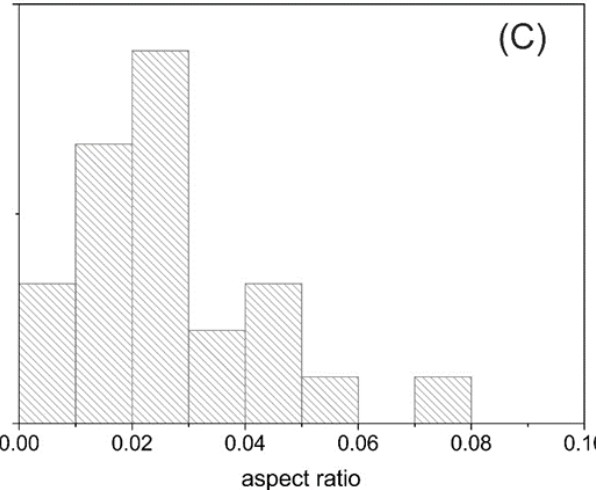


Figure A1: Distributions of aspect ratios of mica grains (A), type I (B) and type II (C) cracks based on analysis
of several RK15-22 thin sections.

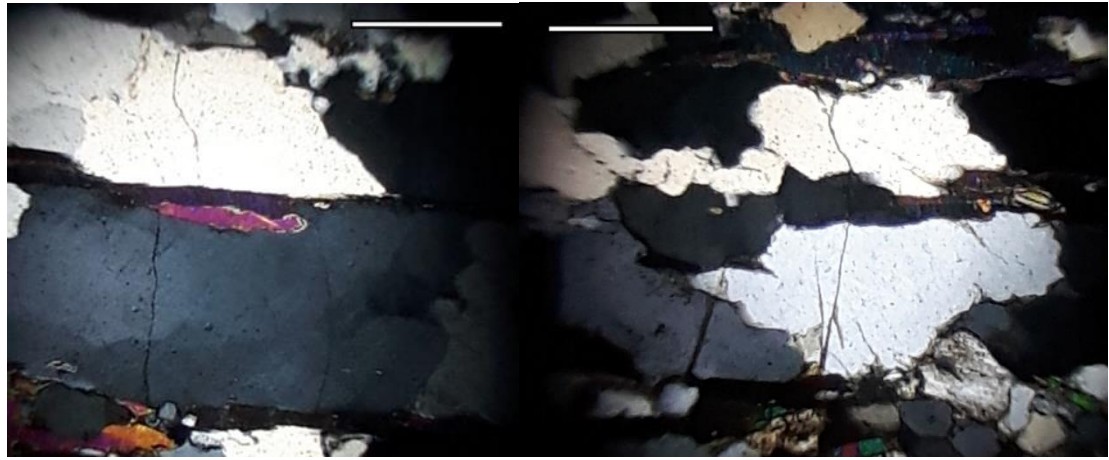


Figure A2: Grains of quartz crossed by type II cracks, XZ plane. Parallel to them are some inclusion trails,
which could be former microcracks sealed by solution precipitation. White scale bar is 0.4 mm.

| Orthogneiss | quartz C-axes | | Paragneiss | quartz C-axes |
|---|---|---|---|---|
| GAN12 | periphery | | GAN08 | periphery |
| JK6 | periphery | | GAN15 | between Z and Y |
| MS17-15 | between Z and Y | | MS17-12B | between Z and Y |
| RK15-9A | periphery | | MS17-12C | between Z and Y |
| RK15-9B | periphery | | RK15-5 | between Z and Y |
| RK15-10 | periphery | | RK15-18 | between Z and Y |
| RK15-11A | between Z and Y | | RK15-22 | between Z and Y |
| RK15-17 | between Z and Y | | RK60 | between Z and Y |
| RK15-20 | periphery | | RK70A | between Z and Y |
| RK15-24B | between Z and Y | | SADU16 | between Z and Y |
| RK15-27B | periphery | | SADU30 | periphery |
| RK15-28 | between Z and Y | | ZAP01 | periphery |
| RK15-30B | between Z and Y | | | |
| RK15-31 | periphery | | | |
| RK63B | between Z and Y | | | |
| RK66 | girdle | | | |
| SADU39 | periphery | | | |


Table A1: CPO patterns of quartz C-axes maxima within the sample set.

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
