# Peer review of "Elastic anisotropies of deformed upper crustal rocks in the Alps"

_Solid Earth, 2021_

## Referee Comment (RC1)

Review Report on Solid Earth MS#: se-2021-27
Title: Elastic anisotropies of deformed upper crustal rocks in the Alps
Authors: R. Keppler, R. Vasin, M. Stipp, T. Lokajícek, M. Petruzálek, N. Froitzheim
Special Issue: New insights into the tectonic evolution of the Alps and the adjacent orogens

Dear Solid Earth Topical Editor M. Vrabec,

Following the instructions provided in your Referee comment request, I present below my report organized in the three sections expected. Rather than introducing corrections and comments in the pdf article file provided (which I have not edited), I list below my viewpoints to help discussion and manuscript improvement, if the authors consider it appropriate.

1. General comments

I have read Keppler and co-workers' article various times with personal interest and attention and, in my opinion, it is adequate for publication in the Solid Earth aforementioned special issue, as it deals with a classic major nappe of the Alps Lepontine dome as well as with the use of sophisticated fabric analysis techniques, derived calculations and rock geophysical measurements applied to metamorphic rocks that record a poly- orogenic geological history. The results of this study may also constitute relevant inputs for ongoing geophysical studies of the Alps (the manuscript mentions the AlpArray high-end seismological array experiments) and in other similar initiatives elsewhere. The text is well written and organized in general, and the figures that illustrate the manuscript are correct and appropriate. For these reasons I would recommend acceptance of the article, though there are some items that, in my opinion, should be clarified and improved previously, all this requiring a minor/moderate revision. I explain below my comments and suggestions in a long section 2 dealing with specific comments and a short section 3 including a few technical corrections.

2. Specific comments

a. "The upper crust within collisional orogens". The authors mention several times in the abstract, conclusions and elsewhere in the MS that they are studying typical lithologies from deformed upper crustal rocks in the Alps. I am mystified by the statement since the rocks dealt with are ortho- and paragneisses that underwent intense ductile deformation coeval with amphibolite facies metamorphism under relatively high pressure conditions (or truly high-P in the eclogite facies). It is not a surprise that the authors describe it in their texts and support it with references. However, what do they actually mean with "upper crustal rocks"? Currently they are in the upper crust, but they did not acquire their principal characteristics in this realm, but at great crustal depth. When the authors design their rock ultrasound velocity experiments at different confining pressures up to 400 MPa they are implicitly admitting at least mid-crustal ambient conditions. The authors should revise what they actually mean and, if necessary, rewrite the parts of the MS where they include the label "upper crustal rocks".

b. On the point of elastic anisotropy measurement and calculations. The authors deal with seismic anisotropy directions in the Alps (with supporting citations) and rock average seismic velocities/anisotropies to support the interest of rare studies as the presented here on gneisses. However, though I can foresee, it is not clear for me (after the current text) which are the specific relationships implied or the eventual applications. On one hand, the authors report on seismic anisotropy parallel or transversal to the orogenic surface trend, but its origin (attending

to the publications cited) is teleseismic wave birrefringence essentially generated in the mantle. The authors mention mineral and stretching lineations of surface metamorphic rocks to support geometrical relationships (that I do not challenge at all), but their contribution to teleseismic signal anisotropy likely is very small due to the relative short tract of crustal segments compared to mantle ones in wave rays, and the still shorter of the uppermost crust (in spite of being much anisotropic). So, which is the contribution expected of crustal rock anisotropy to teleseismic wave anisotropy? Is it relevant? Would it help/hinder seismic lithological reconnaissance along profiles with different orientations? Please, explain. On the other hand, the authors concentrate of the interest of determining "average rock" geophysical properties, but it may be misleading. I wonder if a highly anisotropic medium such as de Adula nappe (with a gentle-dip layered organization of interleaved ortho- and paragneiss units meter to tens or a few hundreds of m thick, and their internal penetrative foliation/lineation fabrics) would be seismically imaged as a coherent unit, nearly transparent in reflection seismology experiments and distinguishable from over- and underlying units in seismic refraction. In the latter case "average rock" properties might be sound. However, the actual lithological organization of the Adula nappe appears to be prone to produce several reflections. In this regard, it should be noted that, rather than the thickness of layers, the key resides in the existence of acoustic impedance contrasts across contacts, which abound between ortho- and paragneisses, and between them and scarcer marble and metabasite lenses, as described in the manuscript. As a conclusion to this item, I consider the authors should describe to what aspect of seismic studies their data and results intend to make a significant contribution.

c. The regional geology of the Adula nappe. The authors present a succinct description of this nappe and illustrate it with a simplified map and a cross section parallel to the regional lineation. In a brief bibliographic survey on the nappe I have found outstanding maps and cross sections that provide a much more precise image of the organization of the unit. I am not sure if I am allowed to include excerpts of them in this report, but in any case they can be accessed, for example, in M. Carvagna-Sani's thesis (2013, Université de Laussanne, "The Adula nappe: stratigraphy, structure and kinematics of an exhumed high-pressure nappe") and related publications on the Adula nappe. They are not cited in this manuscript and, though do not contradict the descriptions presented in this article, they are outstanding as complementary structural support that likely merits citation of consideration to redraw the Fig. 2. Complementary to this, it should be mentioned that the orthogneisses correspond to two groups of different age: Ordovician and Permian. Also, now in relation with the regional geology, in the line 173 (and then in the 281) the authors refer to the "Zapport" phase, which likely is unknown to most readers unfamiliar with Alpine geology. Please, explain what is understood as the "Zapport" phase (pervasive Alpine deformation phase in the northern part of the Adula nappe, associated with the regional foliation, isoclinal folding with approximately N-S fold axes and N-S stretching lineation associating top-to-the-North sense of shear), state its age, and (maybe as a colateral must) that there also exist younger regional phases with local names (Leis and Carassino).

d. Comments on the cross section of the Alps (Fig. 1A), the (over) simplified Fig. 1B and the subdivision of the crust into weakly deformed isotropic upper crust and strongly deformed anisotropic Alpine upper crust (sections 2.1 and 2.2). Apart of the already discussed above on the clarification of the actual meaning of "upper crust", I suggest the authors should reconsider their writings in this section. I understand what the authors intend, but I feel it is an oversimplification not supported by the regional geology as I explain. First, consideration of the crystalline external massifs and the Adriatic basement as recorders of weak or disregarded elastic anisotropy contradicts the fact that these massifs record the imprint of pervasive orogenic deformation, metamorphism and magmatism in internal zones of the Hercynian

orogen (se below some Von Raumer and co-workers' articles on the subject). Also, in several cases the evidence exists that the Hercynian chain reworked an older Cadomian orogen.

Von Raumer, J.F., Stampfli, G.M. and Bussy, F., 2003: Gondwana-derived microcontinents - the constituents of the Variscan and Alpine collisional orogens. Tectonophysics, 365: 7-22.

Von Raumer, J.F., Bussy, F. and Stampfli, G.M., 2009: The Variscan evolution in the external massifs of the Alps and place in their Variscan framework. C.R. Geoscience, 341: 239-252. DOI: 10.1016/j.crte.2008.11.007.

Von Raumer, J.F., Bussy, F., Schaltegger, U., Schulz, B. and Stampfli, G.M., 2013: Pre-Mesozoic Alpine basements – Their place in the European Paleozoic framework. Geological Society of America Bulletin, 125: 89-108. Doi: 10.1130/B30654.

Additionally, during Mesozoic hyperextension in the Alps, several crustal units of distal parts of the involved plate margins (e.g. the Adula nappe basement) underwent widespread ductile deformations that generated foliations, lineations, and elastic mechanical rock anisotropy (there exists also a considerable bibliographic background on this topic that the authors might know, or even articles authored). All this predated the Cenozoic Alpine orogeny, of course, but likely those units are mechanically as anisotropic as those resultant of the Alpine evolution. In my opinion, all this should be acknowledged in the relevant parts of main text, even though actually the authors arrive at their current positions on crustal organization for future modelling.

e. The authors follow a correct linear procedure in explaining how they record mineral rock fabrics and, together with experimental mechanical constant determinations available, follow calculation procedures to quantify ideal rock mechanical properties that then are compared with laboratory measurements in real rocks. This has been done previously in several occasions with peridotites and eclogites and the authors cite the relevant literature grouped into publications dealing with calculations and others dealing with measurements. However, some of the citations contain the two types of information and their appearance in only one group might be misleading. Complementary to this, it is remarkable the contrast between eclogites and gneisses in this regard. Less than five citations relate to gneisses because there exists a real lack of such data. Notwithstanding, there are some references (classical and recent) that might be included in support of the authors' statements (both in the section 2 and in the upcoming). These include the classic works of Christensen, Fountain, Ji and co-workers, such as the listed below (from which a few might be picked) and a recent one on ortho- and paragneiss petrofabrics in a tectonic/metamorphic context similar to that of the Adula nappe (Puelles et al., 2018, J. Metamorph. Geol., 36, 225-254).

Christensen, N.I., 1965. Compressional wave velocities in metamorphic rocks at pressures to 10 kbar. Journal of Geophysical Research, 70, 6147-6164. Doi: 10.1029/JB084iB12p06849.

Christensen, N.I., 1979: Compressional wave velocities in rocks at high temperatures and pressures, critical thermal gradients, and crustal low velocity zones. Jour. Geophys. Res., 84: 6849-6857.

Christensen, N.I. and Fountain, D.M., 1975: Constitution of the lower continental crust based on experimental studies of seismic velocities in granulite. Geol. Soc. Amer. Bull., 86: 227-236.

Christensen, N.I. and Mooney, W.D., 1995: Seismic velocity structure and composition of the continental crust: a global view. Jour. Geophys. Res., 100 B7: 9761-9788.

Fountain, D.M., Arculus, R. and Kay, R.W. (Eds.), 1992: Continental Lower Crust. Elsevier, Amsterdam: 485p.

Ji, S. and Salisbury, M.H., 1993: Shear-wave velocities, anisotropy and splitting in high-grade mylonites. Tectonophysics, 221: 453-473.

Ji, S., Salisbury, M.H. and Hanmer, S., 1993: Petrofabric, P-wave anisotropy and seismic reflectivity of high-grade mylonites. Tectonophysics, 222: 195-226.

Ji, S., Wang, Q. and Xia, B., 2003a. Handbook of Seismic Properties of Minerals, Rocks and Ores. Polytechnic International Press, Montreal, 630 p.

Ji, S., Wang, Q. and Xia, B., 2003b. P-wave velocities of polymineralic rocks: comparison of theory and experiment and test of elastic mixture rules. Tectonophysics, 366, 165-185. Doi: 10.1016/S0040-1951(03)00094-5.

f. Methods. Sample preparation for ultrasonic wave measurement. I am intrigued about how the authors prepared "roughly spherical" rock samples for neutron time-of flight and ultrasonic measurements, as it is not explained in the main text (lines 183-84 and 227). Where they prepared as polyhedra with several facets cut with a discoidal saw, or were they prepared with a spherical grinding machine?

g. Sections 4.5 and 4.6. From my viewpoint, some parts of the texts included in these sections explain methods rather than results and might possibly be reorganized.

h. Theoretical and measured densities used to estimate crack porosity (lines 433-440). In principle the approach to quantify the porosity associated to microcracks may be reasonable, but the 1.7% value appears to me excessive for these rocks. The authors should be aware that considering the actual Ca-Na relationship of gneiss feldspars instead of albite may change the result, as well as would do consideration of the experimental P and S wave data presented to calculate gneiss sample Poisson's ratios, and after it their density under different confining pressures. Since these rocks depart from perfect incompressible materials (with theoretical Poisson's ratios of 0.5), the actual volume changes due to pressurization/decompression should not be ascribed exclusively to microcrack porosity and lowers the 1.7 estimation.

i. Microcracks and the origin of anisotropy at low confining pressure. This is an interesting matter of debate in these rocks and I suggest to include some additional descriptions and points of view. It is out of discussion the relationship between microcracks and seismic velocity magnitude and anisotropy at low confining pressure, as well as the geometrical relationships between microcrack and mesoscopic penetrative structures. In my opinion the microcrack descriptions provided in the manuscript can be improved with the help of the microphotographs presented and, likely, with additional observations. The micrographs presented correspond to standard rock sections normal and parallel to the foliation. In them, in principle would be visible microcracks (intra- and trans-granular) normal to the three principal fabric planes. If the microcracks are narrow rather than wide, it is due to the fact that they form an angle close to 90º with the plane of the image. Alternatively, if they are wide it might be due to either they are open cracks or they are oblique to the image section (actually derived of a 30 μm thick slice) and apparently are wider. These considerations should be borne in mind prior to discussing on crack opening estate. In several microphotographs (and the main text) the authors highlight as microcracks mica exfoliation planes and I do not find convincing evidence in support of that assignation. The reason why those planes are optically individualized by contrast with other cleavage planes can be variable (opaque mineral nanoinclusions, sheared surfaces) and the low mechanical coherence of mica basal planes contributes to it. The authors should provide clear evidence to support those planes are true microcracks. The fact that some mica grains also contain irregular microcracks normal to cleavage planes might support stress states with the plane containing the maximum and minimum stress directions normal to mica cleavage planes. In the case of quartz (notably) the presence of microcracks usually normal to the mineral shape elongation is doubtless and likely record stress relaxation along the lineation direction. No argument against it. However, the Appendix Figure 2 provides clues on the presence of mechanical discontinuities with the same orientation normal to the lineation that parallel to, or laterally grade into fluid inclusion trails. These and other similar features can be thoroughly identified as healed microcracks and are common (though usually overlooked) in quartz-bearing rocks. They denote brittle strain accommodation and immediate crack healing in the presence of fluids under geologically low T conditions, but not as low as those prevailing at the terrain surface. This may be related with

the sentence included in lines 556-558: "It is likely that another system of thinner microcracks is required to match the GMS model and experimental ultrasonic wave velocities in RK15-22 558 at low pressure values" and may be also relevant for the discussions raised in the lines 566-572 and 617-618. There exist a background of scientific articles dealing with these intra-granular penetrative microstructures, published during the last two decades, from which the authors may be aware and that might be taken into account in the discussion section in order to explain the progressive seismic anisotropy decrease and velocity increase until stabilization at significant confining pressures (600-700 MPa, equivalent to crustal depths well above 10-15 km). All this would also apply to discussion on the effect in velocity and anisotropy of other ubiquitous mechanical coherence discontinuities with close geometrical relationships to the macroscpic rock fabric: the grain and subgrain boundaries between identical and different mineral phases. I include below some bibliographic citations that might help.

Derez, T., Pennock, G., Drury, M. and Sintubin, M., 2015. Low-temperature intracrystalline deformation microstructures in quartz. Journal of Structural Geology, 71, 3-23.

Kjøll, H.J., Viola, G., Menegon, L. and Sørensen, B.E., 2015. Brittle-viscous deformation of vein quartz under fluid-rich lower greenschist facies conditions. Solid Earth 6, 681-699.

Palazzin, G., Raimbourg, H., Stünitz, H., Heilbronner, R., Neufeld, K. and Précigout, J., 2018. Evolution in H2O contents during deformation of polycrystalline quartz: an experimental study. Journal of Structural Geology, 114, 95-110.

Raghami, E., Schrank, C. and Kruhl, J.H., 2020. 3D modelling of the effect of thermal-elastic stress on grain-boundary opening in quartz grain aggregates. Tectonophysics, 774, 228242. Doi: 10.1016/j.tecto.2019.228242.

Richter, B., Stünitz, H. and Heilbronner, R., 2018. The brittle-to-viscous transition in polycrystalline quartz: an experimental study. Journal of Structural Geology, 114, 1-21. Doi: 10.1016/j.jsg.2018.06.005.

Schmatz, J. and Urai, J.L., 2011. The interaction of migrating grain boundaries and fluid inclusions in naturally deformed quartz: a case study of a folded and partly recrystallized quartz vein from the Hunsrück Slate, Germany. Journal of Structural Geology, 33, 468-480.

Stünitz, H., Thust, A., Heilbronner, R., Behrens, H., Kilian, R., Tarantola, A. and Fitz Gerald, J.D., 2017. Water redistribution in experimentally deformed natural milky quartz single crystals - Implications for H2O weakening processes. Journal of Geophysical Research, Solid Eath, 122, 866-894. Doi: 10.1002/2016.JB013533.

Tarantola, A., Diamond, L.W. and Stünitz, H., 2010. Modification of fluid inclusions in quartz by deviatoric stress. I: experimentally induced changes in inclusion shapes and microstructure. Contributions to Mineralogy and Petrology, 160, 825-843.

Trepmann, C., Hsu, C., Hentschel, F., Döhler, K., Schneider, C. and Wickmann, V., 2017. Recrystallization of quartz after low-temperature plasticity – the record of stress relaxation below the seismogenic zone. Journal of Structural Geology, 95, 77-92. Doi: 10.1016/j.jsg.2016.12.004.

j. The authors estate in the lines 530-531 that: "The CPO of quartz and mica was not necessarily formed at the same time and could represent different deformation stages", in order to explain apparent discrepancies between CPOs and experimental velocity patterns. Appart of this being difficult to support with the microstructural features described and shown in the Figure 3 (suggestive of coeval mineral fabric development), likely it is unnecessary, bearing in mind the contrasting velocity distribution patterns in the minerals considered (notably quartz and mica), controlled by their crystallography. I suggest removing the sentence.

3. Technical corrections

Line 50. Check the correctness of "is" (are) for anisotropy data.
Lines 53-54. Revise "...can be either be..."
Lines 59, 590, 671 and 726-728. The correct citation year of Ábalos et al. is 2011, not 2010. In the reference list it is wrong, too.

Lines 191-92, 504, 595 and 672. Check the correctness of citing articles submitted to the same journal issue or in preparation (lines 578-579).

Lines 211-212. Avoid one-sentence paragraphs.

Line 220. Check the form of presentation of citations. Should it be "e.g., Vasin et al. (2013),..." instead as (Vasin et al., 2013; ...)?

Lines 237-270. This section describes petrographic data and likely should be presented as an independent section prior to the "Results" section.

Line 253. "kalifeldspar" is used to explain the kfs abbreviation. Is it correct instead of the "Kfs" or "K-feldspar" terms of the usually recommended after Whitney and Evans (2010)?

Lines 239, 256, 380. Here "potassium feldspar" is used, see previous comment.

Line 311. "P-wave anisotropies ... are (instead of is) defined..."

Lines 313, 328, 329, 340, 350, 493, 494, 502, 509, 524, 535 and 653. Add "the" to "... in lineation direction".

Line 646. Add a comma (,) after "collected".

I expect these comments be of help.

Best wishes

B. Ábalos
Bilbao (Spain), April 29, 2021

---

## Author Response (AR1)

**Reviewer 1 Benito Abalos**

1. General comments

I have read Keppler and co-workers' article various times with personal interest and attention and, in my opinion, it is adequate for publication in the Solid Earth aforementioned special issue, as it deals with a classic major nappe of the Alps Lepontine dome as well as with the use of sophisticated fabric analysis techniques, derived calculations and rock geophysical measurements applied to metamorphic rocks that record a poly- orogenic geological history.

The results of this study may also constitute relevant inputs for ongoing geophysical studies of the Alps (the manuscript mentions the AlpArray high-end seismological array experiments) and in other similar initiatives elsewhere. The text is well written and organized in general, and the figures that illustrate the manuscript are correct and appropriate. For these reasons I would recommend acceptance of the article, though there are some items that, in my opinion, should be clarified and improved previously, all this requiring a minor/moderate revision. I explain below my comments and suggestions in a long section 2 dealing with specific comments and a short section 3 including a few technical corrections.

*Thank you very much for this very detailed and elaborate review and the suggestions for additional discussion topics and references. It has largely improved the manuscript. Almost all suggestions have been implemented into the manuscript and explained in the following:*

2. Specific comments

a. "The upper crust within collisional orogens". The authors mention several times in the abstract, conclusions and elsewhere in the MS that they are studying typical lithologies from deformed upper crustal rocks in the Alps. I am mystified by the statement since the rocks dealt with are ortho- and paragneisses that underwent intense ductile deformation coeval with amphibolite facies metamorphism under relatively high pressure conditions (or truly high-P in the eclogite facies). It is not a surprise that the authors describe it in their texts and support it with references. However, what do they actually mean with "upper crustal rocks"? Currently they are in the upper crust, but they did not acquire their principal characteristics in this realm, but at great crustal depth. When the authors design their rock ultrasound velocity experiments at different confining pressures up to 400 MPa they are implicitly admitting at least midcrustal ambient conditions. The authors should revise what they actually mean and, if necessary, rewrite the parts of the MS where they include the label "upper crustal rocks".

*With upper crust we refer to what was originally in an upper crustal position and has an upper crustal composition (granitoids and sedimentary rocks, or ortho- and paragneisses) as opposed to the typically more mafic lower crust. You are right, however, that the rocks investigated here acquired their principal characteristics at much deeper positions. Therefore, we rephrased this at several sections within the manuscript (line 26, 32-35, 114-122, 130 in manuscript with changes tracked). We keep the term "upper crust", when referring to the NFP20 profile as well as our simplified version, as we use it exactly in the sense of Schmid and Kissling (2000). That allows to directly correlate with the NFP20 profile.*

b. On the point of elastic anisotropy measurement and calculations. The authors deal with seismic anisotropy directions in the Alps (with supporting citations) and rock average seismic velocities/anisotropies to support the interest of rare studies as the presented here on gneisses. However, though I can foresee, it is not clear for me (after the current text) which are the specific relationships implied or the eventual applications. On one hand, the authors report on seismic anisotropy parallel or transversal to the orogenic surface trend, but its origin (attending to the publications cited) is teleseismic wave birrefringence essentially generated in the mantle. The authors mention mineral and stretching lineations of surface metamorphic rocks to support geometrical relationships (that I do not challenge at all), but their contribution to teleseismic signal anisotropy likely is very small due to the relative short tract of crustal segments compared to mantle ones in wave rays, and the still shorter of the uppermost crust (in spite of being much anisotropic). So, which is the contribution expected of crustal rock anisotropy to teleseismic wave anisotropy? Is it relevant? Would it help/hinder seismic lithological reconnaissance along profiles with different orientations? Please, explain.

*The contribution of the crustal rock anisotropy to the teleseismic signal is difficult to quantify. Usually, it is neglected because of the small thickness of the crust in comparison to the mantle and because of the lithological variety as well as the large differences in crustal rock anisotropy. In contrast, for the mantle, it is usually assumed that the rock anisotropy is rather constant and hence together with its large thickness most crucial for the overall teleseismic wave anisotropy. However, when crustal thickness is large, due to, for example, stacking in an orogenic wedge, and the anisotropy is high due to strong deformation in a coherent tectonic setting, the contribution of the crustal rock anisotropy to the teleseismic signal could be much more significant (e.g., Levin and Park, 1997; Xu et al., 2007; Huang et al., 2015; Wang et al., 2016). Our data set might be a basis for a better analysis of the teleseismic signal anisotropy and its origin from either crust or mantle sources. This, however, is only one step for an improvement of the teleseismic methods and we are not able to evaluate the progess of teleseismic processing and analysis within the investigations presented here.*

On the other hand, the authors concentrate of the interest of determining "average rock" geophysical properties, but it may be misleading. I wonder if a highly anisotropic medium such as de Adula nappe (with a gentle-dip layered organization of interleaved ortho- and paragneiss units meter to tens or a few hundreds of m thick, and their internal penetrative foliation/lineation fabrics) would be seismically imaged as a coherent unit, nearly transparent in reflection seismology experiments and distinguishable from over- and underlying units in seismic refraction. In the latter case "average rock" properties might be sound. However, the actual lithological organization of the Adula nappe appears to be prone to produce several reflections. In this regard, it should be noted that, rather than the thickness of layers, the key resides in the existence of acoustic impedance contrasts across contacts, which abound between ortho- and paragneisses, and between them and scarcer marble and metabasite lenses, as described in the manuscript. As a conclusion to this item, I consider the authors should describe to what aspect of seismic studies their data and results intend to make a significant contribution.

*We completely agree with this assessment. Our idea of an average crustal rock originates from discussions with seismologists within the Alp Array project (German section 4D-MB). In one of the earlier*

*meetings we presented our data showing different VP patterns and anisotropies within our sample set. This data is generally useful to the seismologists, but some of them asked if we could provide an average value for crustal anisotropy. As a geologist, I fully support the statement that the crust is completely heterogeneous both in composition and in grade of deformation. Even within the same lithology the anisotropy is very variable, as shown by our data. However, the seismologists of 4D-MB are working on an entirely different scale. Some of them need crustal data, just so that it can be subtracted in their models to make a better estimation on seismic anisotropy of the mantle. Average values of the same suit of rocks and of the crust allow for a "best fitting" of their general and large-scale models. This is why in the current study, we present the variety of different crustal rock anisotropies, as well as an anisotropy for an average crustal rock, which of course, from a geologists' point of view is an extreme oversimplification. By presenting both, the range of different anisotropies of various crustal rocks and an average crustal anisotropy, geophysicists working at a crustal scale with higher resolution, as well as those working at a lithospheric scale and taking the crust as one coherent unit have the required data to work with.*

c. The regional geology of the Adula nappe. The authors present a succinct description of this nappe and illustrate it with a simplified map and a cross section parallel to the regional lineation. In a brief bibliographic survey on the nappe I have found outstanding maps and cross sections that provide a much more precise image of the organization of the unit. I am not sure if I am allowed to include excerpts of them in this report, but in any case they can be accessed, for example, in M. Carvagna-Sani's thesis (2013, Université de Laussanne, "The Adula nappe: stratigraphy, structure and kinematics of an exhumed high-pressure nappe") and related publications on the Adula nappe. They are not cited in this manuscript and, though do not contradict the descriptions presented in this article, they are outstanding as complementary structural support that likely merits citation of consideration to redraw the Fig. 2. Complementary to this, it should be mentioned that the orthogneisses correspond to two groups of different age: Ordovician and Permian.

*We now include the suggested references and protolith ages of the orthogneisses found in the Adula Nappe (line 222-223 in manuscript with changes tracked). Furthermore, we updated Figure 2 based on a map by Carvagna-Sani et al. 2014, as suggested. The map is now much more detailed and accurate and we are grateful for this suggestion.*

Also, now in relation with the regional geology, in the line 173 (and then in the 281) the authors refer to the "Zapport" phase, which likely is unknown to most readers unfamiliar with Alpine geology. Please, explain what is understood as the "Zapport" phase (pervasive Alpine deformation phase in the northern part of the Adula nappe, associated with the regional foliation, isoclinal folding with approximately N-S fold axes and N-S stretching lineation associating top-to-the-North sense of shear), state its age, and (maybe as a colateral must) that there also exist younger regional phases with local names (Leis and Carassino).

*We agree. The Zapport phase is regional and not all readers will be familiar with it. More information on the deformation structures formed during the Zapport phase has now been provided in the text (see line 237-240). An age for peak conditions of the UHP metamorphism has been added, which marks the beginning of Zapport phase deformation. As we state in the text, the samples were collected in the*

*central Adula Nappe, where rocks have not been subjected to younger deformation phases, such as the Leiss phase. This is why we prefer not to introduce these in the text.*

d. Comments on the cross section of the Alps (Fig. 1A), the (over) simplified Fig. 1B and the subdivision of the crust into weakly deformed isotropic upper crust and strongly deformed anisotropic Alpine upper crust (sections 2.1 and 2.2). Apart of the already discussed above on the clarification of the actual meaning of "upper crust", I suggest the authors should reconsider their writings in this section. I understand what the authors intend, but I feel it is an oversimplification not supported by the regional geology as I explain. First, consideration of the crystalline external massifs and the Adriatic basement as recorders of weak or disregarded elastic anisotropy contradicts the fact that these massifs record the imprint of pervasive orogenic deformation, metamorphism and magmatism in internal zones of the Hercynian orogen (se below some Von Raumer and co-workers' articles on the subject). Also, in several cases the evidence exists that the Hercynian chain reworked an older Cadomian orogen.

Von Raumer, J.F., Stampfli, G.M. and Bussy, F., 2003: Gondwana-derived microcontinents - the constituents of the Variscan and Alpine collisional orogens. Tectonophysics, 365: 7-22.

Von Raumer, J.F., Bussy, F. and Stampfli, G.M., 2009: The Variscan evolution in the external massifs of the Alps and place in their Variscan framework. C.R. Geoscience, 341: 239-252. DOI: 10.1016/j.crte.2008.11.007.

Von Raumer, J.F., Bussy, F., Schaltegger, U., Schulz, B. and Stampfli, G.M., 2013: Pre-Mesozoic Alpine basements – Their place in the European Paleozoic framework. Geological Society of America Bulletin, 125: 89-108. Doi: 10.1130/B30654.

Additionally, during Mesozoic hyperextension in the Alps, several crustal units of distal parts of the involved plate margins (e.g. the Adula nappe basement) underwent widespread ductile deformations that generated foliations, lineations, and elastic mechanical rock anisotropy (there exists also a considerable bibliographic background on this topic that the authors might know, or even articles authored). All this predated the Cenozoic Alpine orogeny, of course, but likely those units are mechanically as anisotropic as those resultant of the Alpine evolution. In my opinion, all this should be acknowledged in the relevant parts of main text, even though actually the authors arrive at their current positions on crustal organization for future modelling.

*We agree. The influence of structures formed during previous orogenic events has been underrepresented in our text. We now give reference to those and advise that these need to be considered in any crustal scale seismic model of the Alps. In the Adula Nappe, all Mesozoic rifting Hercynic orogeny structures have been overprinted by pervasive Alpine ductile deformation and can therefore be neglected. However, such earlier deformation is well preserved in some other parts of the Alps. For that reason we now also refer to previous deformation and related references (lines 190-194, 201-204).*

e. The authors follow a correct linear procedure in explaining how they record mineral rock fabrics and, together with experimental mechanical constant determinations available, follow calculation

procedures to quantify ideal rock mechanical properties that then are compared with laboratory measurements in real rocks. This has been done previously in several occasions with peridotites and eclogites and the authors cite the relevant literature grouped into publications dealing with calculations and others dealing with measurements. However, some of the citations contain the two types of information and their appearance in only one group might be misleading. Complementary to this, it is remarkable the contrast between eclogites and gneisses in this regard. Less than five citations relate to gneisses because there exists a real lack of such data. Notwithstanding, there are some references (classical and recent) that might be included in support of the authors' statements (both in the section 2 and in the upcoming). These include the classic works of Christensen, Fountain, Ji and co-workers, such as the listed below (from which a few might be picked) and a recent one on ortho- and paragneiss petrofabrics in a tectonic/metamorphic context similar to that of the Adula nappe (Puelles et al., 2018, J. Metamorph. Geol., 36, 225-254).

Christensen, N.I., 1965. Compressional wave velocities in metamorphic rocks at pressures to 10 kbar. Journal of Geophysical Research, 70, 6147-6164. Doi: 10.1029/JB084iB12p06849.

Christensen, N.I., 1979: Compressional wave velocities in rocks at high temperatures and pressures, critical thermal gradients, and crustal low velocity zones. Jour. Geophys. Res., 84: 6849-6857.

Christensen, N.I. and Fountain, D.M., 1975: Constitution of the lower continental crust based on experimental studies of seismic velocities in granulite. Geol. Soc. Amer. Bull., 86: 227-236.

Christensen, N.I. and Mooney, W.D., 1995: Seismic velocity structure and composition of the continental crust: a global view. Jour. Geophys. Res., 100 B7: 9761-9788.

Fountain, D.M., Arculus, R. and Kay, R.W. (Eds.), 1992: Continental Lower Crust. Elsevier, Amsterdam: 485p. Ji, S. and Salisbury, M.H., 1993: Shear-wave velocities, anisotropy and splitting in high-grade mylonites. Tectonophysics, 221: 453-473.

Ji, S., Salisbury, M.H. and Hanmer, S., 1993: Petrofabric, P-wave anisotropy and seismic reflectivity of highgrade mylonites. Tectonophysics, 222: 195-226.

Ji, S., Wang, Q. and Xia, B., 2003a. Handbook of Seismic Properties of Minerals, Rocks and Ores. Polytechnic International Press, Montreal, 630 p. 4

Ji, S., Wang, Q. and Xia, B., 2003b. P-wave velocities of polymineralic rocks: comparison of theory and experiment and test of elastic mixture rules. Tectonophysics, 366, 165-185. Doi: 10.1016/S0040-1951(03)00094

*Thank you for pointing this out. It is quite hard to pick references dedicated to measurements or to modeling only, as majority of works, starting with classical papers, e.g., [Christensen, 1965; Babuška, 1968] already discuss, at least qualitatively, the relation between the measured elastic wave velocities, and elastic properties, mineral composition and preferred orientations of minerals. We tried our best to separate works, which are mostly about ultrasonic measurements, and works, which mostly represent*

*different models of elastic properties of rocks, also paying an attention to papers using both experiments and advanced modeling methods. However, we revised the references in lines 66 - 74 and added the following references:*

*Christensen, N. I.: Compressional wave velocities in metamorphic rocks at pressures to 10 kilobars. Journal of Geophysical Research, 70(24), 6147–6164, 1965.*

*Christensen, N. I.: Compressional wave velocities in rocks at high temperatures and pressures, critical thermal gradients, and crustal low-velocity zones. Journal of Geophysical Research: Solid Earth, 84(B12), 6849–6857, 1979.*

*Christensen, N. I., and Mooney, W. D.: Seismic velocity structure and composition of the continental crust: A global view. Journal of Geophysical Research: Solid Earth, 100(B6), 9761-9788, 1995.*

*Cholach, P.Y., and Schmitt, D.R.: Intrinsic elasticity of a textured transversely isotropic muscovite aggregate: Comparisons to the seismic anisotropy of schists and shales. Journal of Geophysical Research, 111, B09410, 2006.*

*Ji, S., and Salisbury, M.H.: Shear-wave velocities, anisotropy and splitting in high-grade mylonites. Tectonophysics, 221, 453-473, 1993.*

*Ji, S., Salisbury, M.H., Hanmer, S.: Petrofabric, P-wave anisotropy and seismic reflectivity of high-grade tectonites. Tectonophysics, 222, 195-226, 1993.*

*Ji, S., Wang, Q., Xia, B.: P-wave velocities of polymineralic rocks: comparison of theory and experiment and test of elastic mixture rules. Tectonophysics, 366, 165-185, 2003.*

*Puelles, P., Ábalos, B., Gil Ibarguchi, J.I., Rodríguez, J.: Scales of deformation partitioning during exhumation in a continental subduction channel: A petrofabric study of eclogites and gneisses from NW Spain. Journal of Metamorphic Geology, 36(2), 225-254, 2018.*

f. Methods. Sample preparation for ultrasonic wave measurement. I am intrigued about how the authors prepared "roughly spherical" rock samples for neutron time-of flight and ultrasonic measurements, as it is not explained in the main text (lines 183-84 and 227). Where they prepared as polyhedra with several facets cut with a discoidal saw, or were they prepared with a spherical grinding machine?

*For the CPO measurements by neutron diffraction, samples were cut with a saw into roughly equiaxed polyhedral (see figure below) to ensure that the whole sample remains inside the neutron beam during the rotation in the course of experiment, and also to further minimize already small effect of absorption of thermal neutrons in the sample. The samples used in the ultrasonic measurements at increased pressures were prepared as spheres with a high precision (the error on diameter is +- 0.1 mm) using a special machine equipped with two half-spherical cutters.*

[Figure]

g. Sections 4.5 and 4.6. From my viewpoint, some parts of the texts included in these sections explain methods rather than results and might possibly be reorganized.

*We added more details to section 5.5 (formerly 4.6) to highlight all the model assumptions. In addition we added a sentence in the methodical part in lines 292 -293.*

h. Theoretical and measured densities used to estimate crack porosity (lines 433-440). In principle the approach to quantify the porosity associated to microcracks may be reasonable, but the 1.7% value appears to me excessive for these rocks. The authors should be aware that considering the actual Ca-Na relationship of gneiss feldspars instead of albite may change the result, as well as would do consideration of the experimental P and S wave data presented to calculate gneiss sample Poisson's ratios, and after it their density under different confining pressures. Since these rocks depart from perfect incompressible materials (with theoretical Poisson's ratios of 0.5), the actual volume changes due to pressurization/decompression should not be ascribed exclusively to microcrack porosity and lowers the 1.7 estimation.

*Total porosity was calculated based on 1) measured density of the dried sample at ambient conditions; 2) mineral composition of the same sample refined from diffraction data coupled with mineral density values from literature. Indeed, the composition of minerals in the gneiss may differ from that in references. Due to complexity of the rock and insufficient statistics and resolution of measured diffraction patterns, we made no attempt to refine crystal structures of mineral composing the gneiss, which would have allowed us to obtain good estimates of particular mineral densities. Also, mineral volume fractions determined in the diffraction experiment are subject to uncertainties. Therefore, even though 1.7 Vol.% crack porosity may be considered too high for a gneiss, it is a reasonable estimate judging from the experimental data we have. Measured P- and S-wave velocities are not affected, as well as the elastic anisotropy coefficients of the model (since they are ratios). Model velocities are affected in two ways. First, the square root of density enters as a factor for velocities; but here the error is expected to be < 1%, comparable to uncertainty of measured ultrasonic velocities, which is ≈0.5% for P-waves. Second, all crack porosity is attributed to thin microcracks that greatly affect elastic constants. But for rock with thin cracks, linear elastic properties are defined by crack density, which in our case is proportional to the ratio of crack porosity and crack aspect ratio for each microcrack type.*

i. Microcracks and the origin of anisotropy at low confining pressure. This is an interesting matter of debate in these rocks and I suggest to include some additional descriptions and points of view. It is out of discussion the relationship between microcracks and seismic velocity magnitude and anisotropy at low confining pressure, as well as the geometrical relationships between microcrack and mesoscopic penetrative structures. In my opinion the microcrack descriptions provided in the manuscript can be improved with the help of the microphotographs presented and, likely, with additional observations. The micrographs presented correspond to standard rock sections normal and parallel to the foliation. In them, in principle would be visible microcracks (intra- and trans-granular) normal to the three principal fabric planes. If the microcracks are narrow rather than wide, it is due to the fact that they form an angle close to 90º with the plane of the image. Alternatively, if they are wide it might be due to either they are open cracks or they are oblique to the image section (actually derived of a 30 μm thick slice) and apparently are wider. These considerations should be borne in mind prior to discussing on crack opening estate.

*We note that quantification of microcracks is very important, especially for thinner cracks, which greatly affect bulk elastic properties. Despite being much more complex than the often considered Voigt-Reuss-Hill schemes, GMS model still disregards many microstructural features, i.e., representing microcracks as ellipses, with a reasonable, but not exactly measured orientation distribution, without shape distribution, without account for possible local heterogeneities or stress concentrators. It is true that the aspect ratio of microcracks may be overestimated from microscopy on thin sections, due to cracks inclined with respect to the plane, or due to limited resolution. But again, linear bulk elastic properties are defined by crack density. For modelling, we took rather high estimates of crack density and crack aspect ratio, but the model would yield same results for lower crack density of thinner microcracks.*

In several microphotographs (and the main text) the authors highlight as microcracks mica exfoliation planes and I do not find convincing evidence in support of that assignation. The reason why those planes are optically individualized by contrast with other cleavage planes can be variable (opaque mineral nanoinclusions, sheared surfaces) and the low mechanical coherence of mica basal planes contributes to it. The authors should provide clear evidence to support those planes are true microcracks. The fact that some mica grains also contain irregular microcracks normal to cleavage planes might support stress states with the plane containing the maximum and minimum stress directions normal to mica cleavage planes. In the case of quartz (notably) the presence of microcracks usually normal to the mineral shape elongation is doubtless and likely record stress relaxation along the lineation direction. No argument against it. However, the Appendix Figure 2 provides clues on the presence of mechanical discontinuities with the same orientation normal to the lineation that parallel to, or laterally grade into fluid inclusion trails. These and other similar features can be thoroughly identified as healed microcracks and are common (though usually overlooked) in quartz-bearing rocks. They denote brittle strain accommodation and immediate crack healing in the presence of fluids under geologically low T conditions, but not as low as those prevailing at the terrain surface. This may be related with the sentence included in lines 556-558: "It is likely that another system of thinner microcracks is required to match the GMS model and experimental ultrasonic wave velocities in RK15-22 at low pressure values" and may be also relevant for the discussions raised in the lines 566-572 and 617-618. There exist a background of scientific articles dealing with these intragranular penetrative microstructures, published during the last two decades, from which the authors may be aware and that might be taken into account in the discussion section in

order to explain the progressive seismic anisotropy decrease and velocity increase until stabilization at significant confining pressures (600-700 MPa, equivalent to crustal depths well above 10-15 km). All this would also apply to discussion on the effect in velocity and anisotropy of other ubiquitous mechanical coherence discontinuities with close geometrical relationships to the macroscpic rock fabric: the grain and subgrain boundaries between identical and different mineral phases. I include below some bibliographic citations that might help.

Derez, T., Pennock, G., Drury, M. and Sintubin, M., 2015. Low-temperature intracrystalline deformation microstructures in quartz. Journal of Structural Geology, 71, 3-23.

Kjøll, H.J., Viola, G., Menegon, L. and Sørensen, B.E., 2015. Brittle-viscous deformation of vein quartz under fluid-rich lower greenschist facies conditions. Solid Earth 6, 681-699.

Palazzin, G., Raimbourg, H., Stünitz, H., Heilbronner, R., Neufeld, K. and Précigout, J., 2018. Evolution in H2O contents during deformation of polycrystalline quartz: an experimental study. Journal of Structural Geology, 114, 95-110.

Raghami, E., Schrank, C. and Kruhl, J.H., 2020. 3D modelling of the effect of thermal-elastic stress on grainboundary opening in quartz grain aggregates. Tectonophysics, 774, 228242. Doi: 10.1016/j.tecto.2019.228242.

Richter, B., Stünitz, H. and Heilbronner, R., 2018. The brittle-to-viscous transition in polycrystalline quartz: an experimental study. Journal of Structural Geology, 114, 1-21. Doi: 10.1016/j.jsg.2018.06.005.

Schmatz, J. and Urai, J.L., 2011. The interaction of migrating grain boundaries and fluid inclusions in naturally deformed quartz: a case study of a folded and partly recrystallized quartz vein from the Hunsrück Slate, Germany. Journal of Structural Geology, 33, 468-480.

Stünitz, H., Thust, A., Heilbronner, R., Behrens, H., Kilian, R., Tarantola, A. and Fitz Gerald, J.D., 2017. Water redistribution in experimentally deformed natural milky quartz single crystals - Implications for H2O weakening processes. Journal of Geophysical Research, Solid Eath, 122, 866-894. Doi: 10.1002/2016.JB013533.

Tarantola, A., Diamond, L.W. and Stünitz, H., 2010. Modification of fluid inclusions in quartz by deviatoric stress. I: experimentally induced changes in inclusion shapes and microstructure. Contributions to Mineralogy and Petrology, 160, 825-843.

Trepmann, C., Hsu, C., Hentschel, F., Döhler, K., Schneider, C. and Wickmann, V., 2017. Recrystallization of quartz after low-temperature plasticity – the record of stress relaxation below the seismogenic zone. Journal of Structural Geology, 95, 77-92. Doi: 10.1016/j.jsg.2016.12.004.

*Thank you for pointing this out. The possibility of former microcracks sealed by solution precipitation has now been included as a potential additional mechanical discontinuity in our samples. This is now*

*incorporated within in the discussion and some of the mentioned references have been added (see section 6.2. fourth paragraph).*
*Concerning Fig 3: It was not explicitly stated in the manuscript, which microcracks were used in modeling. For self-consistent models, we considered cracks along elongated mica grains. Figure 3 was adjusted to highlight inter- and intragranular cracks instead of mica cleavage traces.*

j. The authors estate in the lines 530-531 that: "The CPO of quartz and mica was not necessarily formed at the same time and could represent different deformation stages", in order to explain apparent discrepancies between CPOs and experimental velocity patterns. Appart of this being difficult to support with the microstructural features described and shown in the Figure 3 (suggestive of coeval mineral fabric development), likely it is unnecessary, bearing in mind the contrasting velocity distribution patterns in the minerals considered (notably quartz and mica), controlled by their crystallography. I suggest removing the sentence.

*The sentence was removed.*

3. Technical corrections
Line 50. Check the correctness of "is" (are) for anisotropy data.
Lines 53-54. Revise "...can be either be..."
Lines 59, 590, 671 and 726-728. The correct citation year of Ábalos et al. is 2011, not 2010. In the reference list it is wrong, too.
Lines 191-92, 504, 595 and 672. Check the correctness of citing articles submitted to the same journal issue or in preparation (lines 578-579).
Lines 211-212. Avoid one-sentence paragraphs.
Line 220. Check the form of presentation of citations. Should it be "e.g., Vasin et al. (2013),..." instead as (Vasin et al., 2013; ...)?
Lines 237-270. This section describes petrographic data and likely should be presented as an independent section prior to the "Results" section.
Line 253. "kalifeldspar" is used to explain the kfs abbreviation. Is it correct instead of the "Kfs" or "K-feldspar" terms of the usually recommended after Whitney and Evans (2010)?
Lines 239, 256, 380. Here "potassium feldspar" is used, see previous comment.
Line 311. "P-wave anisotropies ... are (instead of is) defined..."
Lines 313, 328, 329, 340, 350, 493, 494, 502, 509, 524, 535 and 653. Add "the" to "... in lineation direction".
Line 646. Add a comma (,) after "collected".

*All technical corrections have been applied.*

**Reviewer 2 Sascha Zertani**

General comments:

The manuscript "Elastic anisotropies of deformed upper crustal rocks in the Alps" by Keppler et al., presents a large dataset of TOF neutron diffraction measurements on ortho- and paragneisses from the Adula nappe (Alps). The CPO data is used to calculate petrophysical properties of the rocks, which are compared to ultrasound measurements on two of the samples and modelling of an average composition expected to be representative of the upper crust.

As such the manuscript presents a large dataset on the petrophysical properties of gneisses for which a lack of data exists. The paper is in general well written and figures and tables are appropriate. In my opinion the manuscript will be suitable for SE (and the special issue) though some revisions are necessary. My overall recommendation is moderate revisions.

*We thank you for the detailed review. The additional references improved the discussion as well as the introduction and several misleading remarks have been rephrased and are now more clear to the reader.*

My main concerns are:

The manuscript somewhat misrepresents the advantages and disadvantages of petrophysical properties measured by laboratory measurements compared to those calculated from CPO data, stating insufficient crack closure as the main short coming of laboratory measurements. This is mainly based on two references (Christensen, 1974 and Vasin et al., 2017) and is surely an important aspect to be considered in studies of petrophysical properties. However, it does not capture the bulk of the available literature and it also neglects to mention the simplification made for the calculations based on CPO data (i.e., no cracks, no minor phases, no grain boundaries). This should be treated a bit more openly to capture what the current state of knowledge is.

*Thank you for pointing this out. We now added more details on used assumptions and approximations at the end of section 6.1.2 (formerly 5.1.2)*

Relating to the above point, I am somewhat confused which of the modelled rocks the authors now consider to be the one representative of the crust. Assuming a density of 2.7 g/m³ and lithostatic pressure, 740 MPa corresponds to approximately 28 km, meaning all of the upper crust will be above. At least to me, it is not clear which "average" rock is considered to be representative.

*The average is based on the composition, the CPO and the crack pattern. We model this average rock for different depths, so one can choose input parameters according to the depth of the model. The depth in which we consider the microcracks closed (740 MPa), would be below crustal depth, however rocks of crustal compostion and fabric can still be found at this depth e.g. in subduction zones and collisional orogens. But we fully agree to the criticism, because it was not clear to the reader in the previous text. We now explain this more detail both in section 5.4. and the conclusions.*

The authors claim to upscale and "close the scale gap" (e.g., L97) between the kilometer-scale geophysical studies and the centimeter-scale at which samples are measured. This is of course an important task and not much data exists on the scales in between. However, the manuscript essentially averages some of the phases present in the rocks to construct one "average" rock, which is then considered to be representative. This can be done and is an interesting calculation, but it should be represented as such. The crust does not contain only one rock, as is mentioned numerous times throughout the manuscript. Yet, the authors do not discuss their results in the context of the available scale bridging literature (e.g., Okaya et al., 2019; Facennda et al., 2019; Zertani et al., 2020)

In general, referencing throughout the manuscript is fine, though here and there paragraphs are completely without reference where some are necessary (specifically in the introduction and discussion sections). Some are pointed out in the specific comments below, but I suggest the authors check this again.

*Yes, indeed several important references were missing and have now been added. In addition the results of these previous studies are now elaborated in the introduction (lines 104-111 in manuscript with changes tracked) as well as the discussion (lines 816-818) and are brought into the context of the manuscript.*

Specific comments:

L45-46: It's not really a matter of depth range but of resolution at depth. I suggest rephrasing this sentence including changing "higher depth" to "greater depth".

*Sentence is now rephrased.*

L47: reference for AlpArray initiative missing

*Reference has been added.*

L49: Could you precise what you mean by input parameters? If its petrophysical properties then its redundant and I suggest deleting that part of the sentence

*This part was deleted.*

L52: I suggest changing "lower depth" to "shallower depth"

*Done.*

L50-53: I find this misleading. By no means is the CPO only the main contributor to seismic anisotropy at mantle depth. Neither is everything above the mantle dominated by microfractures. I suggest to be a bit more precise here. Also some references are needed here.

*Additional factors are now mentioned and references have been added.*

L57: suggest changing "single crystal elastic anisotropies" to "single crystal elastic properties"

*Done.*

L59-64: What exactly do you mean by normal depths? Ultrasonic measurements and CPO measurements have been used for decades to deduce petrophysical properties of rocks. For ultrasonic measurements fitting rules exist to obtain crack free velocities (e.g., Ji et al., 2007). Those results obtained from CPO data have other shortcomings: no grain boundaries, no minor phases, no SPO. This sentence should be rephrased.

*We agree that there are other factors influencing elastic anisotropy at depths in which microcracks are closed. This is why we now as suggested, mention these at an earlier point. Therefore we do not repeat this here again.*

L65-70: References needed. What is the information that it is not an issue in the mantle based on?

*Some references have been added and the sentence was rephrased.*

L74: Here would be a good spot to mention what is known about how structural relationships on the km-scale influence bulk petrophysical properties (see references above).

*These points are now included in lines 104-111 and 816-818.*

L76: I would go as far as to claim that there is no such thing as a natural isotropic rock.

*We agree. We rephrased some of section 2.1. to explain that summarizing the isotropic parts in the model is a simplification and more complicated in nature.*

L127: Figure 1B would benefit from some labels: massifs/units, height, ...

*Done.*

L155: I suggest deleting the cross section and rather include a map that shows the sample locations. I would find that much more helpful. Also, please either change or add a universal coordinate system (preferably UTM).

*The cross section gives the reader a better estimate in the thickness and dip of the respective orthogneiss and paragneiss layers so we prefer to leave in in this figure, but we added the sample locations. The Swiss coordinate system is commonly used in publications about the Adula nappe (e.g. Cavargna-Sani et al., 2014 or Nagel et al., 2008), which we prefer this coordinate system for better comparability. However, UTM was added to the figure.*

L160-166: I am not an expert on Alpine geology but I am sure that this information needs some references.

*References were added.*

L175: How are the samples related to the Zapport phase? They do not seem to be eclogite-facies.

*The samples were deformed during the Zapport phase (e.g. typical NS stretching lineation, and do not seem to be deformed by any of the younger deformation phases.*

L206: Which code/software was used for the calculation? Beartex?

*Yes. Name and reference have been added.*

L224-233: Were these measurements performed during loading or unloading of the sample? It is well known that crack closure during loading is to some extent irreversible, which is why such data is often measured during unloading. Specifically with the discussion of this manuscript this is an important information.

*Ultrasonic measurements were done during loading. It is true that there is some irreversible closure of microcracks. For the model, we tried matching ultrasonic wave velocities measured at certain pressure levels by adding certain vol.% of microcracks, without relation to the state of cracks in rock massif. A short discussion is added at the end of section 5.5 (formerly 4.6)*

L244: please be more specific. What signs?

*A more specific description has been added.*

L246: The lines and labels of X and Y direction are hard to see. Also please clarify from which samples these images are. It might also be necessary to provide images of the other samples in the supplementary information.

*The labels for X, Y, and Z directions have been improved and sample names have been added. Since the microstructures of the other samples will be part of another publication, they should not be included in the supplements.*

L251: Table 1: It would be much easier to compare the different mineral assemblages if the minerals were presented in column. I also don't find it particularly helpful to use the Swiss coordinate system. Is there a reason for not using UTM coordinates?

*Columns would require an additional figure. In our opinion, such a figure would not contribute much to the manuscript. As mentioned before, most previous publications about the Adula Nappe use the Swiss coordinate system, so using it makes a comparison to these previous publications easier. We therefore prefer to leave this table as it is.*

L258: "high mica content", please provide a number, e.g., "up to XX vol.%"

*Done.*

L274 and following: I would suggest to provide the names of the samples that the authors are referring to

*This would be a long list for the main text, which is why a table was added in the appendix listing which sample shows which CPO pattern.*

L288-289: This statement should be somehow supported, I suggest to provide all CPO data at least in the supporting information/appendix in order to support the findings.

*The CPO data and microstructures will be part of a separate publication in which the deformation during the Zapport phase will be discussed, which is why we will not include this data in the current manuscript. However, as mentioned before, a list of samples was added in the appendix referring to the specific CPO patterns.*

L302: specify if these are lower or upper hemisphere projections

*This is already specified.*

L387: what is the 5:6 ratio based on?

*The ratio is based on the frequency of occurrence of each CPO pattern in the sample set. We now added this information in the text.*

L548: The authors say that the results from the GMS model and the Voigt model are "quite close". Reuss and Hill averages would likely also be quite close as it is known that V and R become increasingly separated at high anisotropy. I think it is fine to use Voigt averages but considering that the resulting velocities are consistently higher it should be noted here that this is an upper bound (Mainprice and Humbert, 1994).

*We noted that Voigt is an upper boundary of bulk elastic properties. Voigt and Reuss boundaries are in fact the same for the highest possible anisotropy – the single crystal (or single crystal like preferred orientation). And for the random crystallographic texture – meaning isotropic polycrystal – the difference between Voigt and Reuss is maximal [Matthies et al. J. Appl. Cryst. (2001). 34, 585-601]. Though this difference increases with the increase of single crystal anisotropy. Voigt and Reuss averaging schemes provide the same symmetry of elastic properties as self-consistent method.*

L553-554: atmospheric pressure and 2 MPa results should still be shown in Tab. 4.

*Done.*

L565: If this is a "rough estimate" what would the error be on this?

*We rewrote this sentence. "Rough estimate" was not the right phrasing, here.*

L579-581: I am wondering why the marble is included in the manuscript at all since it is not considered to be present in a "significant amount". I would also like to get the authors thoughts on the following: The marble has a fairly high contrast to the more abundant gneisses. Might this not be a reason that even at low abundance it could impact the bulk properties on the km-scale (e.g., Facennda et al., 2019). I do not know the answer but am curious. It might be worthwhile discussing.

*Yes, marble is in fact special in its seismic properties and there is not a large amount of publications on this topic. Discussing this in more detail is a good idea. The discussion of the marble sample is now more elaborate (now chapter 6.1.3.).*

L653: It is not really clear to me what the authors are getting at. Mica is quasi transversely isotropic, which is well known. If mica is the main contributor to anisotropy the bulk rock will have a similar symmetry.

*Yes, this is actually what we say. Mica is likely the main contributor to the anisotropy. But what we measure as stretching lineations in the field is more likely the quartz or feldspar lineation. So when trying to correlate our data to field maps it could be a problem because the tilting of mica around the lineation might not be the same as the quartz or feldspar lineation measured in the field. This is already mentioned in section 6.1.2., which is why we do not provide more detail at this point.*

L673-675: This could be discussed a bit more openly. There is really not that much information on how structural associations influence the bulk signal on such scales available and the topic is surely a matter of debate.

*We amended the text accordingly.*

Technical comments:

L54: change "gained" to "obtained"

L123: suggest to add ", and"

L137: change to "were only weakly over..."

L237: "Sample(s)"; delete "s"

L256: "," after "however"

L258: suggest to change "represent" to "are"

L311: change "anisotropies" to "anisotropy"

L311: I think the authors mean "x100" instead of "100%". Please also specify how Vp-mean is calculated. Is it the mean of all directions or (Vp-max – Vp-min)/2, which is more commonly used.

L333-L342: I don't think that 1.5 sentences require their own subsection. I suggest to combine these. If not than "Metabasites" should be 4.3.4

L423: change "sections" to "section"

L432: change "a following" to "the following" and "." to ":"

L521: change "micaschists" to "mica schists"

L586: suggest to change "determined" to "dominated"

L591: There seems to be a typo in the citation.

L630: add Backus (1962)

L650: aforementioned

L651: has not been well studied

L665: Furthermore

*All technical corrections have been applied.*

References of the literature mentioned in this review not cited in the article are listed below.

Best of luck,

Sascha Zertani

Oslo, July 1st 2021

References:

Faccenda, M., Ferreira, A. M. G., Tisato, N., Lithgow-Bertelloni, C., Stixrude, L., & Pennacchioni, G. (2019). Extrinsic Elastic Anisotropy in a Compositionally Heterogeneous Earth's Mantle. Journal of Geophysical Research: Solid Earth, 124, 1671-1687. https://doi.org//10.1029/2018JB016482

Ji, S., Wang, Q., Marcotte, D., Salisbury, M. H., & Xu, Z. (2007). P wave velocities, anisotropy and hysteresis in ultrahigh-pressure metamorphic rocks as a function of confining pressure. Journal of Geophysical Research: Solid Earth, 112(B9). https://doi.org/10.1029/2006JB004867

Okaya, D., Vel, S. S., Song, W. J., & Johnson, S. E. (2019). Modification of crustal seismic anisotropy by geological structures ("structural geometric anisotropy"). Geosphere, 15(1), 146-170. https://doi.org/10.1130/GES01655.1

**Reviewer 3**

General Comments

In this study, the authors aim to characterize seismic anisotropy in the upper crust of collisional orogens with focus on the European Alps. The goal is to derive representative elastic properties of an average upper crustal rock as it would be measured by seismic observations which are sensitive only for larger scale structures. They select the Adula Nappe as representative unit for upper crustal deformation. Samples of Ortho- and Paragneisses are analyzed for their composition and CPO. CPO and volume percentages are determined in neutron time-of-flight diffractometer. Based on these measurements and considering single crystal anisotropies from laboratory measurements of previous studies, the anisotropy of the samples is calculated showing large variability. An average upper crustal rock is constructed using the volume percentages of relevant mineral phases, their characteristic CPO and average CPO strength. Thin sections and ultrasonic measurements are used to determine characteristic microcrack structure of the samples. This allowed a quantification of the influence of microcracks on the elastic anisotropy of an average upper crustal rock with depth.

The findings are well prepared and the conclusions reasonable based on the presented results. In particular I think this study is significant for the scientific community as it aims to fill the gap in scale between laboratory and seismic measurements, the approach is efficient, as it provides anisotropy for an effective average rock unit and it considers the change of anisotropy with depth by considering microcracks in the samples. I see only some minor issues, which require a minor revision.

*Thank you for this positive and constructive feedback. The additional points, now included in the discussion strongly improved the manuscript.*

Specific Comments

1) The only major issue, I see in the current manuscript, is the fact, that the influence of structural properties like intrusions or layering on the effective anisotropy measured in seismic experiments is mostly neglected throughout the manuscript with two exceptions in the discussion ("layering" first occurs at L477). I would suggest spending some thoughts on this feature at earlier parts of the manuscript and revise some statements in view of this extrinsic anisotropy. Generally, this is not a very big issue, as the authors even provide already an idea of how to deal with this feature in the discussion: L628-630. The following 5 comments relate to the layering issue.

1-1) L37-39: Here, the authors are pointing out the limitations of the "average rock" applicability. I would suggest to also mention, that apparent anisotropy from structural larger-scale features like layering is not considered in the "average rock" approximation and needs additional knowledge, when interpreting seismic data sets.

*We now also advise to consider other structural features in models at this point.*

1-2) L50-64: In the introduction, the authors nicely describe the different features to be considered, when measuring seismic anisotropy at rocks. The authors mention CPO and microcrack influence and the issues about scale differences between sample size, variability, and sensitivity of seismic observations. However, this introduction lacks mentioning structural features as layering, which also produces anisotropy. It should be mentioned, that even isotropic structures produce anisotropy when occurring as intrusions or small-scale layers within larger-scale rock units. I know that this is not considered in the average rock characterization, but this is an important limitation of the applicability, that should be consistently taken care of.

*We agree. This point has been neglected and we now point out additional factors besides CPO and microcracks, which could influence the overall anisotropy of rocks (see lines 44-45, 60-62, 203-204 in manuscript with changes tracked).*

1-3) L167-168: Here, the authors mention the interlayers of lenses from different rock types. If I understood the explanation correctly, these lenses and structures are not considered in the average rock. I agree completely that these structures might be far too small to identify them and their properties in seismic methods. However, I assume that they have a significant effect on the measured anisotropy, which would be a result from the layering of these intrusions in the larger gneiss background rock. I would suggest mentioning this possibility in the discussion and refer to the layering which is identified here.

*This is now mentioned at this point, as well as in chapter 6.3*

1-4) L369-370: Theoretically also heterogeneities like lenses if they occur regularly in the massif would have to be considered. However, the geometry won't be as equally distributed as the properties considered here, therefore I completely agree with the choice of parameters used for the average rock. Still, the statement here reads as if it is a complete list of important factors, which is not completely true.

*We now include a further statement here mentioning heterogeneities that could occur.*

1-5) L579-580: I agree here that the CPO of marble might be of no relevance for the anisotropy of the overall unit. However, the lenses itself with their shape oriented and spread over a wider region might very well produce effective anisotropy in seismic measurements. That depends of course on the vertical and horizontal extent in which these lenses occur. I don't want to say, that the assumption here is wrong. It might very well be, that these lenses have no effect at all. But I would assume, that this very much depends on the geometry and frequency of these lenses as there is a considerable difference in isotropic (P-wave) velocities between marble and gneiss.

*We agree and now mention that depending on the number and the dimensions of these lenses they would have to be considered. We now also point out the further applicability of the anisotropy data of the marble presented, here.*

Not related to layering:

2) L18-19: "This yields difficulties for seismic investigations of tectonic structures at depth since local changes in elastic anisotropy cannot be detected." > Maybe this is not really the point. I would say it is more important, that the diverse and partially strong upper crustal anisotropy might overprint the signal of anisotropic structures at depth, when observed and interpreted in seismic measurements.

*This is a very good point and actually part of the reasons for our work. We included this point in the manuscript.*

3) L46-48: I would suggest to cite the AlpArray seismic network group, here.

(Hetényi, G., Molinari, I., Clinton, J., Bokelmann, G., Bondár, I., Crawford, W. C., Dessa, J.-X., Doubre, C., Friederich, W., Fuchs, F., et al., 2018a. The AlpArray seismic network: a large-scale European experiment to image the Alpine orogen, Surveys in geophysics, 39(5), 1009–1033.)

*This reference was now included.*

4) L103-113: There are a lot more studies looking into the anisotropy of the Alps, which complement the early studies cited here. Below a few recent examples.

Orogen parallel anisotropy:

Bokelmann, G. H. R., Qorbani, E., & Bianchi, I., 2013. Seismic anisotropy and large-scale deformation of the Eastern Alps, Earth and Planetary Science Letters, 383, 1–6, doi: 10.1016/j.epsl.2013.09.019.

Petrescu, L., Pondrelli, S., Salimbeni, S., Faccenda, M., & Group, A. W., 2020. Mantle flow below the central and greater Alpine region: insights from SKS anisotropy analysis at AlpArray and permanent stations, Solid Earth, 11(4), 1275–1290, doi: 10.5194/se-11-1275-2020.

Two-layer anisotropy also from SKS-splitting in the transition to the Eastern Alps (interpreting the two layers as asthenospheric flow above a detached lithospheric slab fragment with frozen in mountain chain parallel CPO):

Qorbani, E., Bianchi, I., & Bokelmann, G., 2015. Slab detachment under the Eastern Alps seen by seismic anisotropy, Earth and Planetary Science Letters, 409(1), 96–108, doi: 10.1016/j.epsl.2014.10.049.

Link, F. & Rümpker, G., 2021. Resolving seismic anisotropy in the lithosphere-asthenosphere in the Central/Eastern Alps beneath the dense SWATH-D network, Front. Earth Sci., provisionally accepted, doi: 10.3389/feart.2021.679887.

*Thank you, these are in fact important references that were missing. We added them to the manuscript.*

5) L114-117: I was a bit irritated by the abrupt change of focus from the general collisional and orogenic setting of the Alps to the very distinct profile, which the measurements and interpretation are based on.

Maybe one or two phrases explaining, why this profile is chosen as representative region for the general setting would be nice, that the reader follows the flow of work here.

*We agree. We now added an explanation as to why we focus on this specific cross section.*

6) L636-638: It would be nice, if there would be some suggestions on the seismic techniques the authors think are suitable to investigate the upper crustal anisotropy. While receiver function techniques are great to infer seismic anisotropy for a certain depth below a single station, they are only sensitive to azimuthal anisotropy. If I look at the foliation/lineation map, I would assume that the radial anisotropy is much larger than the azimuthal anisotropy. Therefore, surface wave techniques or local earthquake tomography might be more suitable, while less accurate in lateral and depth resolution.

*We very much agree that the chosen seismic techniques are crucial for the investigation of the crustal anisotropy. This would require to describe all the different methods and then discuss in detail the pros and cons of them. This cannot be done in a short assessment as it very much depends on the geological setting, the available data as well as the research focus, which the most promising method is. When carrying out this in a comprehensive and reasonable manner we would lose our study focus on structural/microstructural properties that control the crustal anisotropy by lengthy descriptions and discussion on seismic techniques. Therefore, we leave this topic to those who focus on the seismic investigation of the crust and/or the lithosphere.*

Technical comments

L17-18: Doubling of the word "very", maybe use "highly".

L27-28: Insert "the" in "of deformed"

L53-54: Remove the first occurrence of "be"

L220: …rocks are discussed in, e.g., (Vasin et al., 2013; Vasin et al., 2017; Lokajicek et al., 2021).

I would suggest removing the braces "()". "…, e.g., Vasin et al. (2013; 2017) and Lokajicek et al. (2021)."

*All technical corrections have been applied.*